# Comparison of statistical downscaling methods for climate change impact analysis on precipitation-driven drought

Hossein Tabari[1], Santiago Mendoza Paz[1], Daan Buekenhout[1], Patrick Willems[1,2]

[1]KU Leuven, Department of Civil Engineering, Belgium.

[2]Vrije Universiteit Brussel, Department of Hydrology and Hydraulic Engineering, Belgium.

*Correspondence to*: Hossein Tabari (hossein.tabari@kuleuven.be)

**Abstract.** General circulation models (GCMs) are the primary tools to evaluate the possible impacts of climate change; however, their results are coarse in temporal and spatial dimensions. In addition, they often show systematic biases compared to observations. Downscaling and bias correction of climate model outputs is thus required for local applications.

Beside the computationally intensive strategy of dynamical downscaling, statistical downscaling offers a relatively straightforward solution by establishing relationships between small and large scale variables. This study compares four statistical downscaling methods of bias correction (BC), change factor of mean (CFM), quantile perturbation (QP) and event based weather generator (WG) to assess climate change impact on drought by the end of the 21st century (2071-2100) relative to a baseline period of 1971-2000 for the weather station of Uccle located in Belgium. A set of drought related

aspects is analysed: dry day frequency, dry spell duration and total precipitation. The downscaling is applied to a 28-member ensemble of CMIP6 GCMs, each forced by four future scenarios of SSP1-2.6, SSP2-4.5, SSP3-7.0 and SSP5-8.5. A 25-member ensemble of CanESM5 GCM is also used to assess the significance of the climate change signals in comparison to the internal variability of the climate. A performance comparison of the downscaling methods reveals that the QP method outperforms the others in reproducing the magnitude and monthly pattern of the observed indicators. While all methods

show a good agreement on downscaling total precipitation, their results differ quite largely for the frequency and length of dry spells. Using the downscaling methods, dry day frequency is projected to increase significantly in the summer months, with a relative change of up to 19% for SSP5-8.5. At the same time, total precipitation is projected to decrease significantly by up to 33% in these months. Total precipitation also significantly increases in winter, driven by a significant intensification of extreme precipitation rather than a dry day frequency change. Lastly, extreme dry spells are projected to increase in length

by up to 9%.

## 1 Introduction

Our climate system is changing. Since the mid-20*th* century, global warming has been observed (IPCC, 2014). The atmosphere and oceans have warmed, ice and snow volumes have diminished and the sea level has risen. Climate change is linked to a variety of recent weather extremes worldwide. We entered the current decade with Australia's immense bushfires

empowered by severe droughts (Phillips, 2020) and devastating mud slides triggered by extreme precipitation in Brazil (Associated Press, 2020). Nature and human communities all over the world are feeling the impact of global warming, which is projected to become more pronounced in the future (Tabari, 2021). Projections of how global warming will evolve in the coming decades and centuries would be extremely valuable to mankind in order to adapt efficiently.

  Droughts are natural hazards that have an impact on ecological systems and socioeconomic sectors such agriculture,
drinking water supply, waterborne transport, electricity production (hydropower, cooling water) and recreation (Van Loon, 2015; Xie et al., 2018). Quantification of the evolution of droughts on the local level is thus needed to take adequate mitigation measures. The hydrological processes behind drought are complex, with varying spatial and temporal scales. One of the aspects of drought is a lack of precipitation. As the projected increase in total precipitation does not systematically correspond to a decrease in dry days and longest dry spell length (Tabari and Willems, 2018a), beside total precipitation, dry
spells and its building blocks, dry days, should be studied to evaluate the impact of climate change on drought. It is clear that prolonged periods of consecutive dry days can play an important role, for example in replenishing groundwater levels in time for the dry summer season (Raymond et al., 2019).

  Based on observations of more than 5000 rain gauges in the past six decades, Breinl et al. (2020) assessed the historical evolution of dry spells in the USA, Europe and Australia. Both trends towards shorter and longer dry spells were found,
depending on the location. For Europe, extreme dry spells have become shorter in the North (Scandinavia and parts of Germany) and longer in the Netherlands and the central parts of France and Spain. Benestad (2018) also showed that the total area with 24 hrs precipitation between 50°S and 50°N has declined by 7% over the period 1998–2016 using satellite-based Tropical Rain Measurement Mission data. Using climate model data, Raymond et al. (2018, 2019) found a future evolution towards longer dry spells and a larger spatial extent of extreme dry spells in the Mediterranean basin. For Belgium,
Tabari et al. (2015) studied future water availability and drought based on the difference between precipitation and evapotranspiration. Water availability was projected to decrease during summer and to increase during winter, suggesting drier summers and wetter winters in the future.

  General circulation models (GCMs) are the primary tools for climate change impact assessment. However, they produce results at a relatively large temporal and spatial scales, the latter varying between 100 and 300 km, and are often found to
55 show systematic biases in regards to observed data (Takayabu et al., 2016; Ahmed et al., 2019; Song et al., 2020). The bias particularly originates from processes that cannot be captured at the climate model's coarse scales (e.g., convective precipitation). These processes are therefore simplified by means of parametrization, leading to significant bias and uncertainty in the model (Tabari, 2019). In order to work with these results on finer scales, which is usually required for hydrological impact studies, a downscaling approach can be applied. Dynamical downscaling is done by creating regional
climate models that use the output of a GCM as boundary conditions and work at much finer scales (< 50 km). This comes at a large computational cost and does not necessarily account for bias correction (Maraun et al., 2010). An alternative approach is statistical downscaling which derives statistical relationships between predictor(s) and predictand, e.g. the large-

scale historical GCM output and small-scale observations from weather stations and use them to downscale GCM results with relative ease to assess future local climate change impact (Ayar et al., 2016).

To meet the demand of high spatiotemporal results for the hydrological impact analysis of climate change, the use of statistical downscaling methods has recently increased (e.g., Sunyer et al., 2015; Onyutha et al., 2016; Gooré Bi et al., 2017; Smid and Costa, 2018; Van Uytven, 2019; De Niel et al., 2019; Hosseinzadehtalaei et al., 2020). The results of statistical downscaling methods are, nevertheless, often compromised with bias and limitations due to assumptions and approximations made within each method (Trzaska and Schnarr, 2014; Maraun et al., 2015). Some of these assumptions cast doubt on the

reliability of downscaled projections and may limit the suitability of downscaling methods for some applications (Hall, 2014). As there is no single best downscaling method for all applications and regions, though some methods are superior for specific applications, the assumptions that led to the final results for different methods require evaluation. Therefore, end users can select an appropriate method for each application based on the methods' strengths and limitations, the information needs (e.g., desired spatial and temporal resolutions) and the available resources (data, expertise, computing resources and

time-frames).

This study evaluates the assumptions, strengths and weaknesses of four statistical downscaling methods by a climate change impact analysis for the end of the 21st century (2071-2100) relative to a baseline period of 1971-2000. The selected statistical downscaling methods are a bias correction (BC) method, a change factor of mean (CFM) method, a quantile perturbation (QP) method and an event based weather generator (WG). A set of drought related aspects is studied: dry day

frequency, dry spell length and total precipitation. The downscaling is applied to a 28-member ensemble of global climate models, each forced by four Coupled Model Intercomparison Project Phase 6 (CMIP6) climate change scenarios: SSP1-2.6, SSP2-4.5, SSP3-7.0 and SSP5-8.5. The CMIP6 scenarios are an update to the CMIP5 scenarios, called Representative Concentration Pathways (RCPs), that only project future greenhouse gas emissions, expressed as a radiative forcing level in the year 2100 (e.g., RCP8.5). The CMIP6 scenarios link these radiative forcing levels to socioeconomic narratives (e.g.,

demography, land-use, energy use), called Shared Socioeconomic Pathways (SSPs; O'Neill et al., 2015). Historical observations from the Uccle weather station are used for the calibration of the statistical downscaling methods. Two cross-validation methods are applied to evaluate the skill of the downscaling methods. A 25-member ensemble of CanESM5 GCM is also used to test the significance of the climate change signals.

## 2 Data and methodology

### 2.1 Observed and simulated data

The statistical downscaling methods in this study use precipitation time series produced by GCMs as sole predictor. The predictand is also a precipitation time series, but at the local point scale (scale of a weather station). The availability of a long and high-quality time series of observations from the Uccle weather station enables us to effectively calibrate this relationship. The Uccle station is the main weather station of Belgium, located at the heart of the country (Lat = 50.80°, Lon

= 4.35°), and is run by the Royal Meteorological Institute (RMI). Starting in May 1898, the precipitation is being recorded at 10-min intervals with the same instrument, making it one of the longest high-frequency observation time series in the world (Demarée, 2003). In this study, the 10-min observations are aggregated into daily precipitation values, the same temporal scale as the considered GCMs. The information lost by this aggregation is of low interest for studying drought.

Small samples are subject to "the law of small numbers" (Kahneman, 2012) and can provide misleading results due to
their high sensitivity to the presence of strong random statistical fluctuations (Benestad et al., 2017a, b; Hosseinzadehtalaei et al., 2017). To obtain more robust results, daily precipitation simulations for the historical period 1971-2000 and the future period 2071-2100 from a large ensemble of 28 CMIP6 GCMs are used in this study (Table 1). The data for the grid cell covering Uccle is selected for every GCM using the nearest neighbour algorithm. To give the GCMs in the ensemble an equal weight in the analysis, the one run per model (1R1M) strategy (Tabari et al., 2019) is applied. For one of the GCMs
(CanESM5), 25 runs (r1 – r25) are considered in order to allow for quantification of the internal variability in GCM output. To allow for intercomparison of possible futures, multiple scenarios are selected. The four Tier 1 scenarios in ScenarioMIP (CMIP6) are chosen. This set of scenarios covers a wide range of uncertainties in future greenhouse gas forcings coupled to the corresponding socioeconomic developments (O'Neill et al., 2016). On a practical note, the GCM runs for these four scenarios are widely available since they are a basic requirement for participation in CMIP6.

**2.2 Statistical downscaling methods**

Four statistical downscaling methods were selected for this study based on their complexity and the way they treat dry spells. Each method has a different take on the downscaling of dry spells. This study aims at examining the influence of these factors in the statistical downscaling using four methods which are different in methodology and complexity. While BC and CFM are considered simple and computationally fast and straightforward methods that do not modify dry spells in
downscaling, QP and WB are more advanced methods that adjust dry spells. BC applies a bias correction to the selected statistics, whereas the other three downscaling methods return a modified precipitation time series. BC utilizes a direct downscaling strategy by applying the relative change factors directly to the dry spell related research indicators. The other three methods opt for an indirect downscaling strategy towards dry spells by integrating the changes in dry days, which are downscaled directly into a coherent time series. For this, CFM solely relies on the temporal (precipitation) structure present
in the GCM time series. QP on the other hand is expected to actively favour clustering of dry days. Lastly, WB makes use of a probability distribution to sample dry events from. While the precipitation change factor methods (BC, CFM and QP) assume independency between successive wet days and apply changes at the daily time scale, which can be problematic when successive wet days are part of a longer lasting event, WG identifies precipitation events and applies the same change factor to all precipitation within that event.

## 2.2.1 Bias correction (BC) of statistics

The first statistical downscaling method applies a bias correction to the statistics that describe the precipitation time series. Consequently, this method does not return a precipitation time series, unlike the three other downscaling methods. This method can be regarded as a BC method applied directly to statistics (indicators) instead of to a daily precipitation time series. The BC factor is calculated as the ratio of the observed indicator to the model indicator of historical simulations, and then applied on the model indicator of scenario simulations to derive projected indicator. The indicators used in this study, to which the BC is applied, are discussed later on in subsection 2.4.

An important assumption of all BC methods is that the climate model precipitation bias is time-invariant, which might not be the case (Leander and Buishand, 2007). Furthermore, BC methods assume the temporal structure of wet and dry days of the scenario-projected precipitation by the climate model is accurate. Successive days are also assumed independent.

## 2.2.2 Change factor of mean (CFM) method

The change factor of mean method or delta change method is frequently applied in the literature. The same simple rationale of the BC method can be applied by using a change factor approach instead. Here, no correction is applied to GCM precipitation projections. Instead, the relative change between the historical and scenario simulations of the GCM is used to calculate a change factor that can then be applied to the observed time series (Sunyer et al., 2012, 2015). The method applied to the precipitation $P$ of day $t$ in month $m$ can be summarized by Eq. 1.

$$P_{m,t}^{Proj} = a_m . P_{m,t}^{Obs} \tag{1.a}$$

$$\text{in which } a_m = \frac{P_{m,\dots}^{GCMScen}}{P_{m,\dots}^{GCMHis}} \tag{1.b}$$

In this notation, the precipitation is given for month $m$ and time step $t$ in the observations ($Obs$), and $GCM_{Scen}$ and $GCM_{His}$ refer to the scenario and historical simulations of GCMs, respectively. For this implementation, the change factor is calculated per month.

CFM does not change the number of dry days directly. However, since the change factor is applied to all precipitation in a given month, days in the Uccle time series with precipitation values close to the wet day threshold (dry day: $P < 1.0$ mm, wet day: $P \geq 1.0$ mm) can change state, depending on the change factor $a_m$. The Uccle precipitation time series has a resolution of 0.1 mm. The wet days nearest to the threshold have a value of 1.0 mm, while the closest dry days have a value of 0.9 mm. Consequently, wet days are changed into dry days for $a_m < 1.0$, while a transformation of dry days into wet days requires $a_m > \frac{1.0}{0.9} = 1.11$. In conclusion, CFM is expected to show slight changes in terms of dry days, with a bias towards rising the number of dry days, and thus the dry spells they compose. The mean monthly total precipitation changes projected in this method can be used as a reference for the other methods.

An important assumption made in all CF methods is that the changes at local (weather station) level are the same as the changes described at the spatial, grid-averaged scale of climate models. Different from the BC methods, the CF methods

assume the temporal structure of the observed time series is preserved. Furthermore, it is assumed in the CFM method that all precipitation in a given period (i.e. month or season) is changed by the same factor, regardless of the time step considered or the precipitation intensity observed. In addition, the method assumes consecutive days are independent.

### 2.2.3 Quantile perturbation (QP) method

QP methods form a more advanced approach to the application of change factors. The core principle of the methods is that the change factors are calculated and allocated based on the exceedance probability of the precipitation intensities. More precisely, the observed daily precipitation with exceedance probability $p$ is modified by a change factor obtained by comparing the scenario and historical simulations of climate models for the same exceedance probability $p$. This is opposed to the idea of applying the same change factor to observed precipitation amounts ranging from zero to the most extreme

values, as is done in CFM.

In the QP version applied by Ntegeka et al. (2014) is used here in which the empirical exceedance probabilities $p_k$ are estimated by making use of the formula $(\frac{k}{n+1})$ for Weibull plotting positions, where $k$ is the quantile rank (1 for the highest) and $n$ is the number of wet days. This approach can change the exceedance probabilities strongly in comparison to the linear interpolation of the cumulative density function represented by $(\frac{k}{n})$ especially for extreme ranks. This approach was shown to

be best suited for estimating return periods of extreme events (Makkonen, 2006).

In QP, the dry day frequency is perturbed by making use of a two-step perturbation process. In a first step, change factors are calculated to determine the relative change in dry day frequency between the scenario and historical simulations of climate models. These determine whether dry days in a given month should be converted to wet days or the other way around. This is done randomly using a stochastic approach. However, an assumption concerning the clustering of dry days is

made: only wet days preceded or followed by a dry day are eligible for the conversion or only dry days both preceded and followed by a wet day can be converted. After the wet/dry day perturbation step, the precipitation intensity of remaining wet days is perturbed by change factors derived from comparing the scenario and historical simulations of climate models. Due to the randomness introduced by the dry day perturbation step, multiple time series are generated. A sensitivity analysis is executed by varying the number of simulations (see Text S2 and Figs. S1-S3). The selection of the 'best' simulation is based

on four indicators that can be derived from a precipitation time series: the mean (M), coefficient of variability (CV), skewness (S) and average monthly autocorrelation coefficient for a lag of 1 day ($\rho_1$). Using these four indicators, the distance $D$ between the climate change signals of the generated series and the GCM time series, for a given month $m$, is calculated:

$$D_m = \sum_{i=1}^{4} \left( \frac{I_{i,m}^g}{I_{i,m}^{Obs}} - \frac{I_{i,m}^{GCMScen}}{I_{i,m}^{GCMHis}} \right)^2 \qquad (2)$$

where $g$ denotes the generated series and *Obs*, *GCM_Scen* and *GCM_His* have the same meaning as in Eq. 1. The simulation corresponding to the smallest distance is selected as the best one.

The CF assumptions remain in place for the QP method, as well as the assumption regarding consecutive days as independent. Unlike the CF method, it is now assumed that extreme and non-extreme precipitation amounts can change with different factors. The temporal structure of the observed time series is not explicitly changed. Furthermore, it is assumed that the highest relative changes are applied to the days with the highest daily precipitation. The method allows for an explicit perturbation of the temporal structure of the observed time series.

### 2.2.4 Event based weather generator (WG)

The fourth selected statistical downscaling method in this study is the stochastic and event based approach developed by Thorndahl et al. (2017) which is not directly based on change factors but generates stochastic time series instead. Consequently, it belongs to the category of the weather generators. The method constructs a stochastic time series by alternating wet and dry events. Wet events are sampled from an observed point precipitation time series. Observed dry event durations are fitted to a two-component mixed exponential distribution (three parameters: $\lambda_a$, $\lambda_b$ and $p_a$ in Eq. 3) from which dry events durations (also called inter-event durations, $t_{ie}$) are sampled. Both sampling operations are performed for each season separately.

$$f(t_{ie}) = p_a\big[\lambda_{a,ie}\exp(-\lambda_{a,ie}t_{ie})\big] + (1 - p_a)\big[\lambda_{a,ie}\exp(-\lambda_{b,ie}t_{ie})\big] \tag{3}$$

where $\lambda_{a,ie}$ and $\lambda_{b,ie}$ are the rate parameters for two populations, $a$ and $b$, with different exponential distributions and $p_a$ is the weight of population $a$. More information about the two-component mixed exponential distribution can be found in the supplementary information (Text S1). A popular way to fit this type of distribution to data points is by application of iterative Expectation-Maximization algorithms (Yilmaz et al., 2015). An implementation of this algorithm for fitting mixed exponential distributions is included in the R package 'Renext' (Deville and IRSN, 2016).

Figure 1 shows the two-component mixed exponential density functions that are fitted to the empirical probabilities of observed (Uccle) dry event lengths. The fitted distributions underestimate the proportion of inter-events with a duration of 1 day. This underestimation is countered when sampling since the complete range [0, 1.5] of sampled durations is rounded to 1 day. Figure 2 shows the two-component mixed exponential cumulative density functions that are fitted to the seasonal empirical cumulative density functions of observed (Uccle) daily precipitation intensities. The fitted distributions are very close to simple exponential distributions since $p_a \approx p_b \approx 0.5$ ($p_b = [1 - p_a]$) and $\lambda_a \approx \lambda_b$.

When the sampling processes are performed, the three parameters ($\lambda_a$, $\lambda_b$ and $p_a$) of the two-component mixed exponential distribution are converted into stochastic variables (sampled from a uniform distribution) in order to accommodate for climate change. A similar approach is used for extreme precipitation, requiring the sampling of two parameters. In total, five parameters are sampled from uniform distributions for each season.

The stochastic nature of this method requires a large number of simulations. These are evaluated using several target variables and the corresponding change factors, which are calculated using the GCM ensemble. For each climate change scenario, one simulation is picked from the accepted simulations as the 'best' simulation, based on the performance it shows

for different target variables. This method requires to make an arbitrary choice on several parameters: the boundaries of the

uniform sampling intervals, the number of simulations, target variables and their weights. The sampling boundaries for the dry spell parameters and the number of simulations are the subject of a sensitivity analysis (see Text S2 and Fig. S4). The other parameters are further discussed in detail hereafter.

*Parameters for precipitation change factor function*: The two parameters (slope $\alpha$ and intercept $\beta$) of a linear change factor function (Eq. 4), used to alter event precipitation amounts in function of its exceedance probability, are sampled from

uniform distributions.

$$c(i) = \alpha F(i) + \beta \tag{4}$$

in which

$$F(i) = p[1 - \exp(\lambda_a i)] + (1 - p)[1 - exp(\lambda_b i)] \tag{5}$$

where $c(i)$ is the change factor as a function of intensity $i$, $F(i)$ *is* the probability of a given rainfall intensity $i$ being less

than or equal to $i$ using the same two-component mixed exponential distribution used for fitting the inter-event durations (Eq. 3), $\lambda_a$ and $\lambda_b$ are the rate parameters for populations $a$ and $b$ with different exponential distributions and $p_a$ is the weight of population $a$.

Thorndahl et al. (2017) specify that the sampling boundaries are empirically selected by executing the method for very broad sampling ranges and iteratively narrowing them down based on the simulations that are accepted. When applying this

strategy, a test run comprising 50,000 simulations did, however, not show clear boundaries for these parameters. Instead, sampling ranges are chosen at $0.000 - 0.050$ and $0.80 - 1.20$ for $\alpha$ and $\beta$, respectively, for all seasons. These values correspond well to the parameter ranges found by Thorndahl et al. (2017) for the accepted runs in their study.

*Target variables*: The performance of a simulation is evaluated based on a set of target variables. The target values for these variables are determined by application of change factors to the corresponding variables of the observed time series $H$.

The value for target value $i$ for simulation $j$ is denoted as $M_{i,j}$ and the climate change factor for target value $i$ as $cf_i$. The performance $P$ is then calculated using Eq. 6a. Assuming a Gaussian distribution of the target variables, the acceptance criterion $P_{crit}$ for each target variable is taken as its 95% confidence interval (Eq. 6b). A simulated time series $j$ is accepted when $P_{i,j} > P_{crit,i}$ for all target variables $i$.

$$P_{i,j} = 1 - \frac{|cf_i \cdot H_i - M_{i,j}|}{cf_i \cdot H_i} \tag{6a}$$

$$P_{crit,i} = 1 - \frac{2 \cdot \sigma_{cf,i}}{cf_i} \tag{6b}$$

For all accepted simulations, the overall performance is calculated as a weighed sum of all individual target variable performances. For $n$ target variables and weights $w_i$, this becomes $P_j = \sum_{i=1}^{n} w_i P_{i,j}$.

The set of target variables in the original implementation is altered in order to fit the specific needs of this study better. Two target variables related to precipitation with T = 2 years and T = 5 years are removed. Instead, five new target variables

are added, assuring the annual and seasonal number of dry days is adequately reproduced in the accepted simulations (Table 2). The weights, attributed to each target variable for calculation of the overall performance, are attributed in favour of the

dry days target variables in order to reflect their importance for this study. The largest weights are assigned to the target variables that are expected to undergo the largest changes, which are expected to be the hardest to simulate.

Like the other statistical downscaling methods, some assumptions are made in the WG method. It makes assumptions
similar to change factor methods due to the selection procedure. The changes found for climate model grid-averaged spatial scales are treated as targets for the stochastic simulations. Furthermore, this weather generator assumes wet event durations will not change, while dry event durations will. In addition, it is assumed that observed time steps with larger precipitation amounts will have a relatively larger increase in precipitation in comparison to time steps with lower precipitation amounts.

## 2.3 Validation of statistical downscaling methods

All downscaling methods are prone to errors and require a proper validation (Benestad, 2016). We validate the four downscaling methods to assess how they reproduce dry day frequency, dry spell duration and total precipitation. An observation-based cross-validation is applied to evaluate the skill of CFM, QP and WG in terms of the relative error metric. As the BC method cannot be validated based on the observation-based cross-validation, it is evaluated using an inter-model cross-validation (Räty et al., 2014; Schmith et al., 2021). In the observation-based cross-validation, also called holdout
method (Piani et al., 2010, Dosio and Paruolo, 2011) and perfect predictor experiment (Maraun et al., 2019a, b), observations are regarded as being pseudo-climate model data. The validation period is defined from 1971 to 2000 same as the historical period of the GCMs. As the dominant modes of internal variability in mid-latitudes have cycles of several decades (Schlesinger and Ramankutty, 1994; Tabari and Willems, 2018b), a large temporal distance between calibration and validation periods is required to acquire stable approximations of forced changes (Maraun and Widmann, 2018). A period in
the far past (1900-1929; the first 30-year period in Uccle observations) is thus selected as the calibration period.

In the inter-model cross-validation, each of the 28 GCMs employed in this study are by turns considered as pseudo-observations. The historical simulation (1971-2000) of the pseudo-observations (verifying GCM) is used for the calibration of the remaining GCMs (projecting GCMs), and the scenario simulation (2071-2100) of the verifying GCM is utilized for the validation of projecting GCMs. The relative error for each indicator is computed as the absolute difference between the
projected indicator from projecting GCMs and the validation indicator from the verifying GCM for the end of the 21st century (2071-2100) divided by the validation indicator. For the 28 GCMs ($N = 28$), 756 combinations ($N \times [N − 1]$) are obtained to validate the BC method, also providing confidence intervals for the relative error.

## 2.4 Research indicators

In order to compare climate change scenarios and statistical downscaling methods, five types of research indicators are used
in this study (Table 3). The most important indicators for this study are related to dry days, dry spells and total precipitation. A typical threshold used for separating wet and dry days is 0.1 mm (Pérez-Sánchez et al., 2018; Breinl et al., 2020). This value corresponds to the standard resolution used for precipitation observations. However, in recent climate change projection studies this threshold is often chosen higher, at 1 mm (Raymond et al., 2018; Tabari and Willems, 2018a; Kendon

et al., 2019; Han et al., 2019). This is done to counter the tendency of coarse climate models (GCMs) to overestimate the
285 number of days with low precipitation (Tabari and Willems, 2018a), the so-called 'drizzle problem' (Moon et al., 2018).

Following the definition used in the climate change study by Raymond et al. (2018), a dry spell is defined as consecutive dry days with less than 1 mm of precipitation. Furthermore, they define several classes of dry spell lengths (Table 4), based on the percentiles of dry spell length calculated using the historical period of the study. Dry spells are not to be confused with the terms *dry events* (Willems, 2013; Willems and Vrac, 2011) or *inter-events* (Sørup et al., 2017; Thorndahl et al., 290 2017) used in the statistical downscaling methods. This is due to the definition of dry spells comprising *consecutive* dry days ($\geq 2$ days). In the discussed method implementations, dry events and inter-events respectively have minimum lengths of one day and even shorter than one day.

The number of dry days is considered on a monthly basis. To assess changes in dry spell patterns, the classification discussed in the literature review by Raymond et al. (2018) is followed. For each of the five classes based on dry spell 295 lengths, the number of dry spells is calculated. An additional indicator gives more information on the class containing the longest dry spells, *very long dry spells*. Here, the mean length of very long dry spells is used as an indicator. The indicators related to dry spells are calculated over the entire 30-year period to prevent splitting dry spells up. The last indicator used in this research for drought assessment is the mean monthly precipitation.

An additional precipitation indicator describes the extreme precipitation in a given month *m* and allows for a rough 300 comparison in terms of extreme precipitation, which is useful to compare how the different statistical downscaling methods handle extreme precipitation. This indicator is defined as the monthly maximum daily precipitation averaged over the 30-year period.

## 2.5 Significance testing of climate change signals

The projected research indicators found after statistical downscaling can be compared to those found in the observed time 305 series. For research indicator *i* with value *I*, this climate change signal (CCS$_i$) is defined as $I_i^{Proj}$ divided by $I_i^{Obs}$. Something can be said about the significance of the projected CCS in the GCM ensemble by comparing it with the internal variability of one climate model. A significance test is executed based on the Z-score (Tabari et al., 2019). Here, the stochastic variable X represents CCS$_i$. The null hypothesis of the Z-test corresponds to a situation without climate change: the mean of CCS$_i$ is equal to 1 ($H_0: \mu = 1$). The standard deviation $\sigma$ can be estimated by the standard deviation of CCS$_i$ found over the 25 310 CanESM5 runs, denoted $s_{i,25}$. The difference between these GCM runs is that they are initialized using different starting conditions, i.e. points in the pre-industrial control run. The differences in CCS for these 25 runs can thus be attributed to the internal variability of the climate system, which is regarded as 'noise'. Consequently, the CCS is said to be significant if the *signal-to-noise* ratio (S2N), here equal to $|Z|$, is sufficiently large. Similar as in Tabari et al. (2019), the Z-test is applied to the median CCS$_i$ over the 28-member GCM ensemble. For a confidence level of 95%, the null hypothesis is rejected if

$|Z| = \Phi\left(1 - \frac{0.05}{2}\right) > 1.96$. 10% and 20% significance levels correspond to $Z = 1.64$ and $1.28$, respectively. An important

assumption in this approach is that $s_{i,25}$ is a representative description for all climate models within the GCM ensemble.

## 3 Results

Before using the statistical downscaling methods for projecting the drought related indicators, their skill is validated in terms of the relative error metric (Figs. 3 and 4). For total precipitation, QP and CFM with a relative error of $< 4\%$ for different

320 months outperform WG and BC. The distribution of the relative error for total precipitation adjusted by BC is generally shifted towards higher values for higher level scenarios. For the number of dry days, QP with a relative error of $\simeq 1\%$ for all months is clearly the best performed method, followed by WG for January to May and by either WG or CFM for the remaining months. BC is the worst method for the number of dry days, for which the relative error increases with scenario level. As for dry spells, QP can be considered the best method for the number of very short to large dry spells. The difference

between the skills of the four methods is small for the number of very short and short dry spells, while it gets bigger as the spells become longer. For all the methods, the relative error enlarges for longer spells.

Once the downscaling methods are evaluated, the future projections for the drought-related indicators are derived from the methods. Figure 5 shows the projections for the number of dry days per month with and without statistical downscaling. The results are characterized by the median of the CMIP6 GCM ensemble and the changes can be seen by comparing the

330 projected indicator and the observed one at Uccle station. For BC, CFM and QP, each member of the ensemble is downscaled separately. As a consequence, the variation within the downscaled ensemble can also be looked at. This is not possible for WG since it downscales the ensemble as a whole. The median indicator values for BC, CFM and QP show a similar pattern. Across the four scenarios, the number of dry days increases between June and September in comparison to the Uccle observations. As expected, the increase becomes larger for higher level scenarios. The number of dry days remains

about the same for the other months. WG projects a lower number of dry days during the summer months. The inter-model variation for dry day number projections tends to be the largest for BC, closely followed by QP. CFM shows a considerably smaller inter-model variation. The results for the CMIP6 GCMs without downscaling differ quite largely from the downscaled series during the winter months, and the difference gets smaller towards summer.

To analyse the dry spell related indicators, dry spells are categorized by the quantiles of dry spell lengths in the observed

(Uccle) time series. Table 4 gives an overview of the limits for each dry spell class. The projections for the number of dry spell indicators (the number per class, over a 30-year period) are shown in Fig. 6. Results not only vary strongly between statistical downscaling methods, but also between CMIP6 scenarios. The results generally point to an increase in the number of medium, large and very large dry spells in comparison to the observations. The magnitude of the changes is found to increase with scenario level for all the methods except WG which shows no clear pattern. Even the sign of the WG derived

changes for extreme lengths of dry spells (very short and very long) alters between positive and negative among scenarios. The increase in the number of medium, large and very large dry spells for BC and CFM is in the expense of a decrease in the

number of short and very short dry spells. Without downscaling, the CMIP6 GCMs generally show a lower number of dry spells than the downscaled results across all classes and a higher value for the dry spell length indicator. Next to very long dry spell, dry spell length (mean length of very long dry spells), which is a characteristic of the most extreme dry spells, is also analysed (Fig. 6). In comparison to the historical observations, the general trend is towards an increase in dry spell length. The magnitude of the increase in dry spell length rises with scenario level. The inter-model spread of the number and the length of dry spells for the methods follows a similar pattern to the number of dry days, which is large, medium and small spreads for BC, QP and CFM, respectively.

The results for mean monthly precipitation are given in Fig. 7. Compared to the historical situation, the clearest changes appear in the summer months (June – September) where precipitation decreases according to all methods except WG. WG shows a decrease between June and August for higher-end scenarios (SSP3-7.0 and SSP5-8.5). Between October and May, BC, CFM and QP projections show a precipitation increase, although less pronounced than the decrease in the summer months. In terms of the inter-model variability, BC, CFM and QP show a similar spread. The CMIP6 GCM ensemble without downscaling indications higher values for winter season and lower values for summer season in comparison to the downscaled series.

The second series of research indicators related to precipitation is monthly maximum daily precipitation. As mentioned earlier, this research indicator does not attribute towards the drought investigation that is the main objective of this study. Rather, this indicator is used to gain further insight in the way the selected statistical downscaling methods work, as many statistical downscaling methods are originally developed for extreme precipitation studies. The maximum daily precipitation on a monthly basis and averaged over the 30-year period, is given in Fig. 8. An interesting observation is that the downscaling methods project a very similar and relatively slight increase during winter season, while for summer season the results vary greatly. The largest changes in comparison to the historical period are given by CFM, where a considerable decrease is found during the summer months. The results of WG are again less similar to the results of the other downscaling methods in terms of the change magnitude, while it provides the same change direction. When comparing the CMIP6 GCM projections before and after downscaling, they shows relatively similar results during the winter months, while the downscaled projections for the summer months are lower.

The assessment of the significance of the results is based on the relative changes in comparison to the historical observations. In this study, this relative change is defined as the climate change signal. The median climate change signal of the GCM ensemble is given in Tables 5 and 6 for the different scenarios, statistical downscaling methods and research indicators. Based on the variation in climate change signals within the 25 CanESM5 runs (after downscaling), the significance of the median climate change signal of the ensemble can also be indicated. This is not possible for WG as it does not downscale each member of the ensemble separately. The number of dry days and total precipitation mainly show significance for the medium to high level scenarios during the summer months and to a lesser extent during the winter months. There is an agreement between the downscaling methods for the significance of the changes for the number of dry days and total precipitation. The significance of the changes in maximum daily precipitation is only found for CFM during

the summer months and for all methods in December. Summer precipitation extremes in Belgium are, however, convective in nature which are not well represented by coarse-resolution GCMs (Kendon et al., 2017), necessitating the use of convection-permitting climate models (grid spacing of $\leq 4$ km) for their simulations (Tabari et al., 2016). The changes in dry spell length are significant for CFM and QP under almost all scenarios, while none of the BC derived changes are statistically significant. In contrast, BC is the method with the largest number of significant changes in the number of dry spells. That is, all changes in the number of medium and long dry spells for all scenarios obtained from BC are significant. The changes in these classes of dry spell number for higher level scenarios are also significant by CFM. While the QP derived changes for these classes are not significant, QP identifies some significant changes in extreme classes (very small and very long) of dry spells.

## 4 Discussion

### 4.1 Statistical downscaling methods

From the results, it is clear that the statistical downscaling methods can act quite differently. By uncovering where these differences stem from, the performance of the statistical downscaling methods for drought research can be quantified. Hence, the results for the four statistical downscaling methods are discussed and linked to the methods' strengths and weaknesses.

### 4.1.1 BC method

The first method, BC, applies a bias correction directly to the research indicators. This means no underlying time series is created. A first consequence is that not all projections are necessarily compatible with each other if the indicators are interdependent. This is the case for the number of dry spells since there are only a limited number of dry days to be distributed over the different classes of dry spells.

Second, the number of extreme events such as long and very long dry spell is limited. In the 30-year period of observations in Uccle, only 20 and 11 long and very long dry spells occurred, respectively, while the number of these events varies substantially among CMIP6 projections (15 – 100 and 11 – 88 under SSP5-8.5). This leads to very large bias correction factors which in turn lead to (over)spectacular results after downscaling (see Fig. 6). The same problem holds true for the dry spell length indicator. An absolute bias correction approach instead of a relative one might be more appropriate. In the same spirit, Raymond et al. (2019) discuss changes in extreme dry spell lengths in absolute terms (days) rather than percentages.

Note that these concerns do not take away from this method's ability to qualitatively downscale indicators such as number of dry days or total precipitation. These indicators are often projected by making use of relative change factors, as is also the case for the other statistical downscaling methods.

### 4.1.2 CFM method

CFM does not account directly for changes in the number of dry days. This CFM method applies a change factor to the observed time series in order to match the changes in total precipitation. For this specific research indicator, the result should consequently be no different than the one obtained using BC. The slight differences between these methods in Fig. 7 might be attributed to rounding differences.

The rationale behind the application of this method to assess changes in drought finds its roots in the definition of the dry day threshold at 1 mm. As mentioned earlier, this is done to counter for the so-called 'drizzle problem' GCMs are affected by, meaning they overestimate the number of days with low numbers of precipitation. Consequently, days with precipitation amounts just below this threshold are classified as 'dry', while they might very well be lifted above this threshold in months where total precipitation is increased by the statistical downscaling method. Inversely, the wet days with precipitation just over the limit might convert to dry days in months with a decreasing total precipitation. Fig. 7 shows this effect quite clearly for the summer months, where total precipitation is projected to decrease. The relative change in the number of dry days under SSP5-8.5 scenario (+5.5% in August) remains, however, rather small in comparison to BC (+17.5%) or QP (+14.7%), which both account for the number of dry days directly. The relative error of the drought-related indicators obtained here for CFM is far smaller than that reported for extreme precipitation in the Mid-Europe region (Schmith et al., 2021).

The most interesting aspect in applying CFM, however, is the lack of vital assumptions as to how changing the number of dry days affect the dry spells. All required information is contained within the time series created by the GCM. In this light, the general trends for the number of dry days and dry spell length indicators as projected by QP are interesting to examine, while keeping in mind that the underlying changes in the number of dry days are considerably smaller than one would find through a direct change factor approach.

### 4.1.3 QP method

An important aspect for drought assessment in QP method is in the form of the separate dry day perturbation step. Here, the time series is perturbed to match the projections of the number of dry days. Consequently, QP method should be equal to BC in terms of the number of dry days projections. This is not exactly true, as shown in Fig. 5, but the differences are small enough to attribute them to rounding off the results differently. As dry days are the building blocks of dry spells, a solid downscaling approach towards the number of dry days is vital for downscaling the number of dry spells and dry spell length. Out of the four methods considered in this study, QP is the best performed method in downscaling the number of dry days.

### 4.1.4 WG method

In several ways, WG seems to be the odd one out among the considered statistical downscaling methods. The original implementation of this method (Thorndahl et al., 2017) does, for instance, not downscale each member of the CMIP6 GCM ensemble separately as is the case for the other methods. Instead, WG aims to create one time series that corresponds well to

the mean of the ensemble, at least in terms of the selected target variables. In theory, an implementation that downscales each member of the GCM ensemble separately is possible. Tests executed in this direction uncovered a practical problem related to the sampling boundaries for parameters governing the dry event duration distribution. As shown in the sensitivity analysis (see Text S2), WG struggles to deal with large changes in the number of dry days, e.g. under SSP5-8.5. While the changes in the sensitivity analysis are averaged out over the GCM ensemble, they are not when downscaling each ensemble member separately. The much larger changes that would have to be tackled by the WG would require much larger sampling boundaries. The largest change found in the GCM ensemble (one of the CanESM5 runs under SSP5-8.5) is a decrease of 40% in the number of dry days. To accommodate for this change, sampling boundaries upwards of 70% are required in theory. It is expected that an even larger sampling range is needed, in combination with large numbers of simulations to generate a comfortable number of accepted simulations. Testing at 40% and 30,000 simulations showed that for many members in the GCM ensemble, no accepted simulations could be generated. This is especially true for the SSP5-8.5 scenario.

For the monthly indicators, the number of dry days and total precipitation, BC, CFM and QP more or less match the temporal structure found in the Uccle observations. This is not however the case for WG. Two reasons can be identified for this. First, the method is implemented on a seasonal basis, following the original implementation (Thorndahl et al., 2017). Therefore, the method does not try to match changes in the number of dry days or total precipitation for every month but rather for the season as a whole. A comparison between a seasonal and a monthly implementation might be interesting to further investigate this method. A monthly implementation is expected to require larger numbers of simulations in order to achieve similar numbers of accepted simulations. This is due to the larger number of research indicators present (monthly instead of seasonal). Second, the downscaled time series do not necessarily match the mean of the GCM ensemble exactly for each research indicator. On the contrary, the method accepts all simulated time series that remain within the maximum deviation for each target variable (Table 7). These maximum deviations can be very large, e.g. $\simeq 48\%$ for extreme precipitation and $\simeq 15\%$ for total precipitation in summer (both under SSP5-8.5). Consequently, simulations that are far from the mean projections for some of the key research indicators (e.g. number of dry days) enter into the pool of accepted simulations and might be selected as the 'best' simulation due to the high performance of the simulation for other target variables. This explains the difference of WG for the number of dry days (Fig. 5) and total precipitation (Fig. 7) in comparison to the downscaling methods that accurately downscale these indicators, even when grouping the results per season (DJF – MAM – JJA – SON).

The inaccurate simulation of the number of dry days affects the dry spell related indicators. It was concluded earlier that this is also the case for CFM. An additional concern for this downscaling method is that only one data point (best simulation) is available for comparison in Fig. 6, instead of the 28 data points (size of the ensemble) for the other downscaling methods. While this concern also holds true for the other indicators, it is mitigated by using these indicators (or similar) as target variables. In order to prevent the problems encountered with a relative bias correction applied directly to the dry spell indicators (see BC), this strategy cannot be followed for dry spell related indicators.

## 4.2 Significance of climate change signals

The significance of the results is initially introduced to evaluate how the signal (median climate change signal) compares to the noise present in the CMIP6 GCM output, before downscaling. These results are implicitly formulated in Table 5 since they are the same as the BC results. As discussed earlier, only a limited number of research indicators are found to be significant, even at a relatively low significance level of 20%. The main takeaway from these results is that the increasing number of dry days (up to 19% for SSP5-8.5) and the decreasing total precipitation (up to 33% for SSP5-8.5) in the summer months are found to be significant. Total precipitation in January and December also significantly increases due to a significant increase in precipitation intensity as the changes in the number of dry days (or wet days) are not significant. Furthermore, a significant lengthening of dry spells up to 9% and a significant increase in the number of medium and larger dry spells as high as 90% are found. Our results suggest wetter winters and drier summers for Belgium, consistent with the results obtained from the CMIP5 GCMs (Tabari et al., 2015). An increase in the length of extreme dry spells (Breinl et al., 2020) and in aridity conditions (Tabari, 2020) was also found for west Europe.

The same methodology is followed to assess the significance of the results after downscaling. From the discussion on the different downscaling methods, it is clear that not all indicators are necessarily downscaled accurately. The results should thus be interpreted with care. As mentioned earlier, the main concern for BC is the direct downscaling of the dry spell related indicators, due to the small sample size and the lack of coherence between the projections for the different dry spell classes. As a consequence, the 90% increase for long dry spell is interpreted as an inaccurate result rather than a significant one. For CFM, it is observed that total precipitation is downscaled most accurately. The significant results for maximum daily precipitation during the summer months should thus be considered as inaccurate. QP on the other hand shows some interesting results. This method downscales the monthly indicators (number of dry days, total precipitation and maximum precipitation) accurately. Dry spells are not downscaled directly, but by randomly integrating the number of dry days changes in the original time series. This assures the dry spell related indicators are coherent. As such, the significant 8.7% increase at the 5% level for dry spell length under SSP5-8.5 is the most interesting results across all downscaling methods.

## 4.3 Research indicators

Five different types of research indicators are selected for this research. This subsection shortly evaluates the value of these indicators for this research.

The number of dry days and total precipitation are both straightforward indicators that are widely used in literature for drought assessment (e.g., Tabari and Willems, 2018a; Hänsel et al., 2019). Both have proven to be useful to compare statistical downscaling methods (e.g., Ali et al., 2019) and gain insight in these methods since they often rely directly on them. For example, CFM is governed solely by total precipitation while WG directly considers number of dry days and QP method both through its target variables. In this study, both indicators were structured on a monthly basis. It is believed that a seasonal structure could also form a successful alternative.

As for the dry spell indicators, the number of dry spells related indicators offer interesting insights into the changes that occur within the dry spell household. The system introduced by Raymond et al. (2018) offers a straightforward but decent classification. Beside the different dry spell class indicators, the dry spell length indicator is introduced in order to gain further insight into the longest and most important dry spell class and fulfils this role adequately. An indicator describing the most extreme dry spell within the 30-year period could make for an interesting addition in future research.

Last is the maximum daily precipitation per month averaged over the 30-year period. This indicator does not capture all nuances of extreme precipitation, but gives a rough impression of extreme precipitation changes. In this research, the maximum daily precipitation indicator merely functions as a simple illustration on how the statistical downscaling methods process extreme precipitation differently. It is not a relevant indicator for drought research.

## 5 Conclusions and recommendations

Four statistical downscaling methods were applied to the CMIP6 GCM ensemble for climate change impact assessment on drought. The main difference is how they treat the downscaling of dry spells. BC uses a bias correction applied directly to the dry spell research indicators, while the other downscaling methods approach dry spell downscaling indirectly by changing dry day frequency in the precipitation time series. CFM uses the information available in the time series ('drizzle') to convert the state (wet or dry) of days that are just below or over the wet day threshold (1 mm/day). QP applies changes in dry day frequency at random places in the time series. WG samples dry event lengths from a mixed exponential distribution. Other indicators, the number of dry days and total precipitation are downscaled directly across all methods, except for CFM which only takes total precipitation into account.

The results for BC mirror the relative changes found in the CMIP6 GCM ensemble. While this seems to be a good approach for the number of dry days and total precipitation, the dry spell related indicators seem to be inflated due to the relative change applied to indicators with low occurrences, e.g. only 11 dry spells with a length over 25 days are observed in the Uccle precipitation time series. CFM fails to project the number of dry days correctly. While this might have been expected as the number of dry days is not taken into account during downscaling, this method is tested to see what dry spell patterns are 'hidden' into the original time series. Due to the poor projections of dry day frequency, this method is not fit to evaluate dry spell changes.

Similar to BC, QP downscales the number of dry days directly using the change factors found in the CMIP6 GCM ensemble. By altering the time series at random to match the dry day frequency, the dry spells are altered indirectly. Out of the four statistical downscaling methods used in this study, QP has the overall best performance in reproducing the magnitude and monthly pattern of the observed indicators. Lastly, the event based weather generator (WG) is a complex but potent method. This method uses the relative changes found in the CMIP6 GCM ensemble as targets for the number of dry days and total precipitation. A rather large deviation from these projections is, however, allowed. This results in a poor downscaling of the changes in dry day frequency and consequently in dry spells, despite the interesting approach it offers

towards dry spells (mixed exponential distribution). Stricter selection criteria and more optimized target variables should improve this method's performance, likely at a larger computational cost.

Considering the significance of the changes and the consistency among the downscaling methods, dry day frequency significantly increases in the summer months by up to 19% for SSP5-8.5. This dry day frequency increase may lead to a total precipitation decrease by up to 33%, as precipitation intensity remains unchanged or insignificantly decreases. Total precipitation is also projected to significantly increase in the winter months, as a result of a significant intensification of extreme precipitation. Furthermore, extreme dry spells are projected to be longer by up to 9%.

WG offers ample opportunity for further improvement. The method could be structured per month instead of per season to capture month-to-month variation to match the other methods. Application of the method to each GCM in the ensemble would create more data points, allowing the quantification of the significance of the results found by using this method. Furthermore, alterations could be made to the acceptance criterion in order to lower the allowed deviations from the changes projected by the GCMs. This is especially important for accurate simulations of the number of dry days. With the same goal in mind, the mix of target variables and their corresponding weights could be changed (e.g., only target variables related to dry days). Furthermore, different dry event duration distributions (e.g., Weibull, exponential, gamma, generalized Pareto) can be considered beside the mixed exponential distribution that is used in this research.

There is also room for new downscaling methods that are optimized to deal with dry spells. For example, a method that uses quantile mapping to assess dry spell changes (similar to precipitation downscaling in the QP method) could make for an interesting comparison to the other methods. In addition, a method that applies absolute changes to the dry spell indicators could be studied. The probabilities of dry spells such as the parameters of the probability density function (PDF) can also be downscaled. Because the statistics of dry spell lengths tend to follow a binomial distribution (Wilby et al., 1998; Semenov et al., 1998; Wilks, 1999; Mathlouthi & Lebdi, 2009), probability $p$ that it rains on a specific day is estimated as $p = 1/\text{mean}$ spell length. Similar method was used for the downscaling of heatwaves in India (Benestad et al., 2018).

Several research indicators can be used to assess the statistical downscaling methods for the impact analysis of climate change on drought. In combination with total precipitation (water supply), one could consider evapotranspiration (water demand) to assess dryness (Greve et al., 2019; Tabari, 2020) and water availability (Tabari et al., 2015; Konapala et al., 2020). Furthermore, additional indicators can be used to study dry spells. Beside the mean length of very long dry spells, the maximum dry spell length over a certain period can also be of interest. Furthermore, the temporal behaviour of dry spells could be studied, for example based on their starting, ending or middle day. This might be especially useful to assess the impact of dry spells during the wet season, when water tables have to be replenished in order to bridge the dry summer season.

*Data availability.* CMIP6 GCM data used in the study are freely available at the ESGF website (https://esgf-index1.ceda.ac.uk). The Uccle historical precipitation time series were provided by the Royal Meteorological Institute (RMI) of Belgium (www.meteo.be)

*Author contribution.* All authors collaboratively conceived the idea and conceptualized the methodology. SMP and DB carried out the analysis. HT wrote the initial draft of the paper. All authors discussed the results and edited the paper.

*Competing interests.* The authors declare that they have no conflict of interest.

*Acknowledgements.* The first author thanks the Research Foundation – Flanders (FWO) for financial support (grant number: 12P3219N).

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

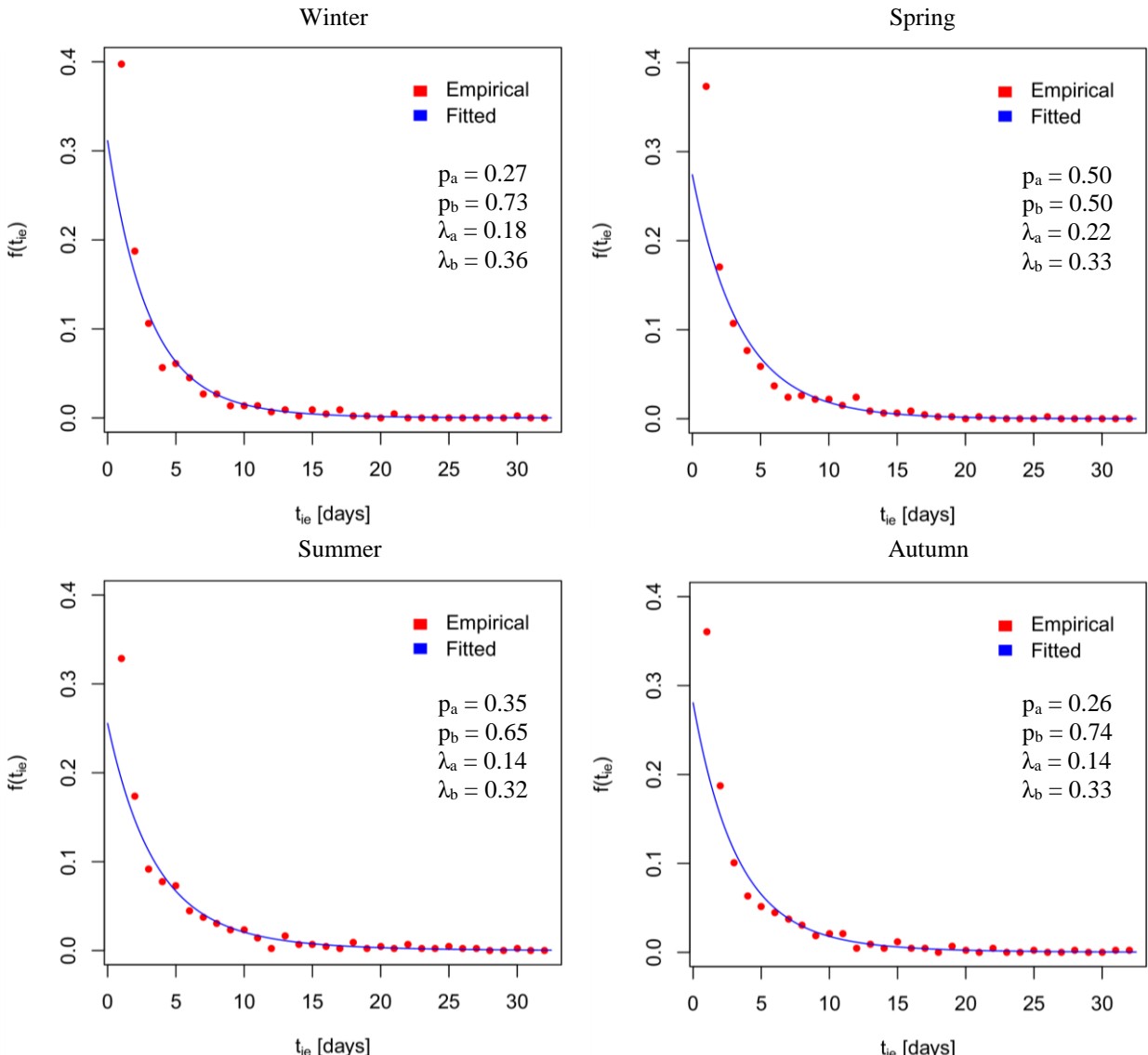

**Figure 1.** Fitted two-component mixed exponential distributions to seasonal empirical probabilities of observed (Uccle) dry event durations. $\lambda_{a,ie}$ and $\lambda_{b,ie}$ are the rate parameters for populations $a$ and $b$ with different exponential distributions, $p_a$ is the weight of population $a$ and $p_b$ is the complement of the weight of population $a$ ($p_b = [1 - p_a]$).

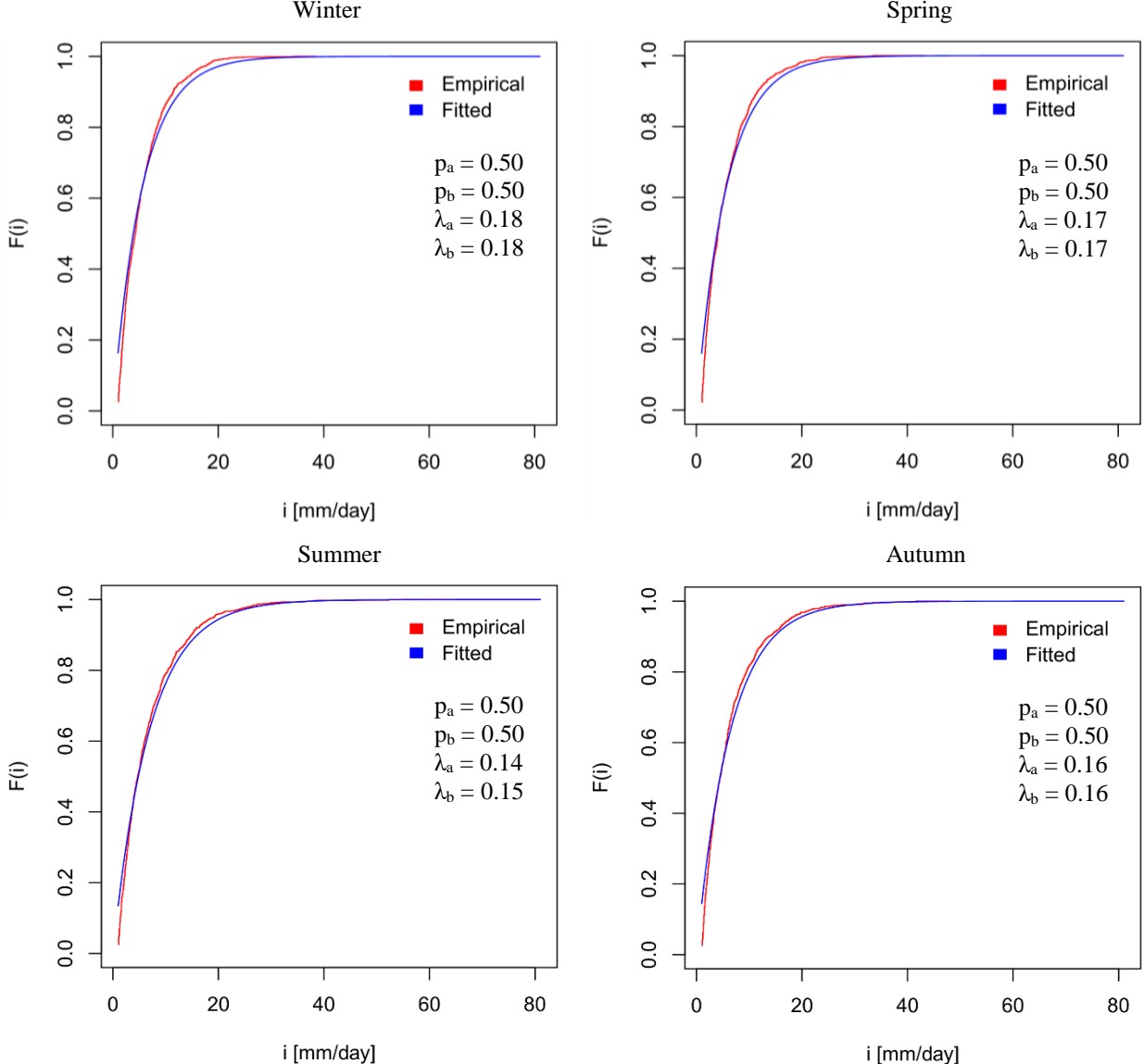

**Figure 2.** Fitted two-component mixed exponential distributions compared to empirical cumulative density functions of seasonal observed (Uccle) daily precipitation intensities. $\lambda_{a,ie}$ and $\lambda_{b,ie}$ are the rate parameters for populations $a$ and $b$ with different exponential distributions, $p_a$ is the weight of population $a$ and $p_b$ is the complement of the weight of population $a$ ($p_b = [1 - p_a]$).

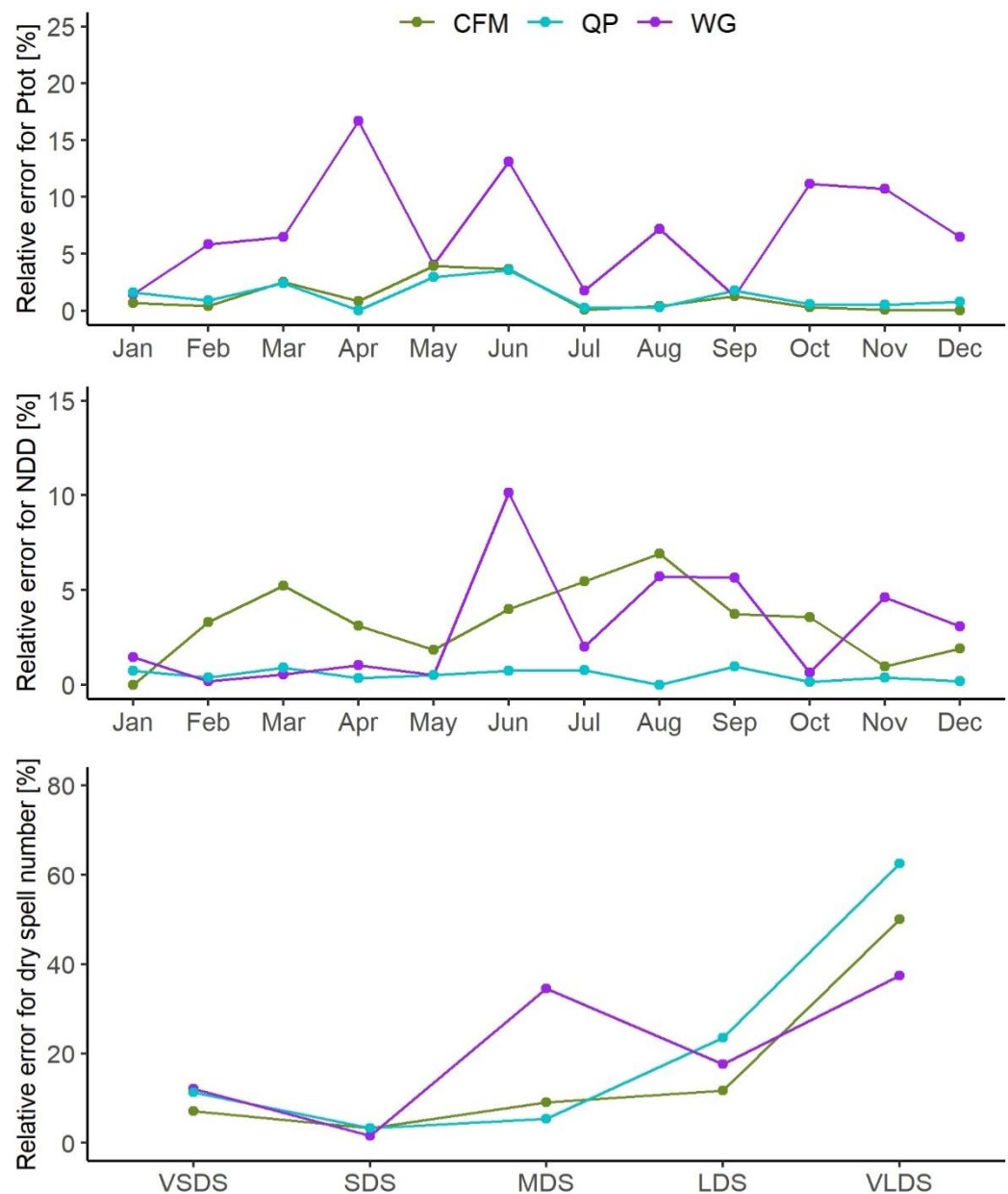

**Figure 3.** Relative error of the observation-based cross-validation for the drought-related indicators (Ptot: monthly precipitation, NDD: number of dry days, dry spell number). Each colour represents a downscaling method. VSDS, SDS, MDS, LDS and VLDS denote very short, short, medium, long and very long dry spells defined in Table 4, respectively.

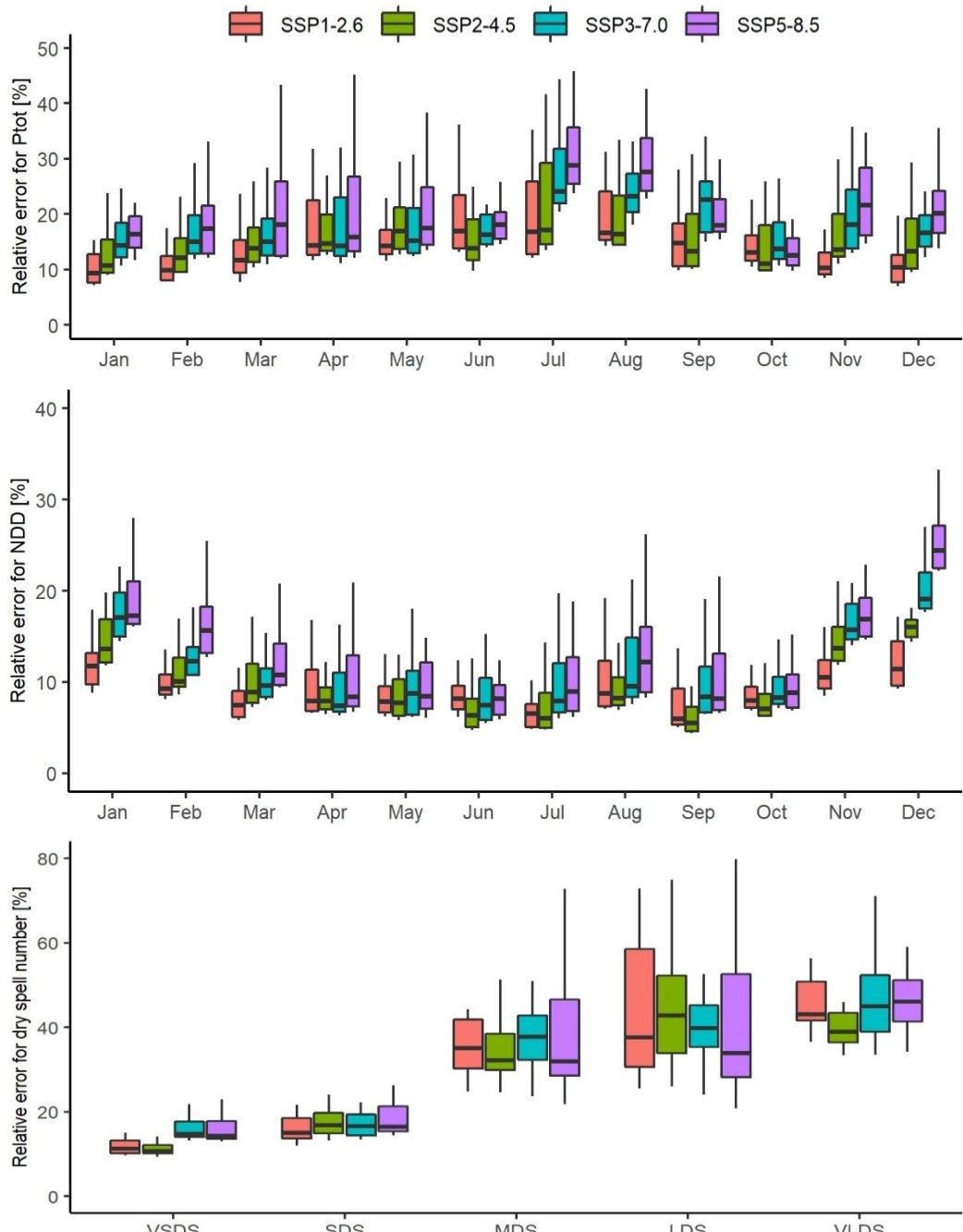

**Figure 4.** Relative error of the inter-model cross-validation of the BC method for the drought-related indicators (Ptot: monthly precipitation, NDD: number of dry days, dry spell number). VSDS, SDS, MDS, LDS and VLDS denote very short, short, medium, long and very long dry spells defined in Table 4, respectively. Top and bottom of the box show the 75th and 25th percentiles of the relative error, respectively. Top and bottom of the whiskers show the 5th and 95th percentiles, respectively. Horizontal black line in the middle of the box represents the median.

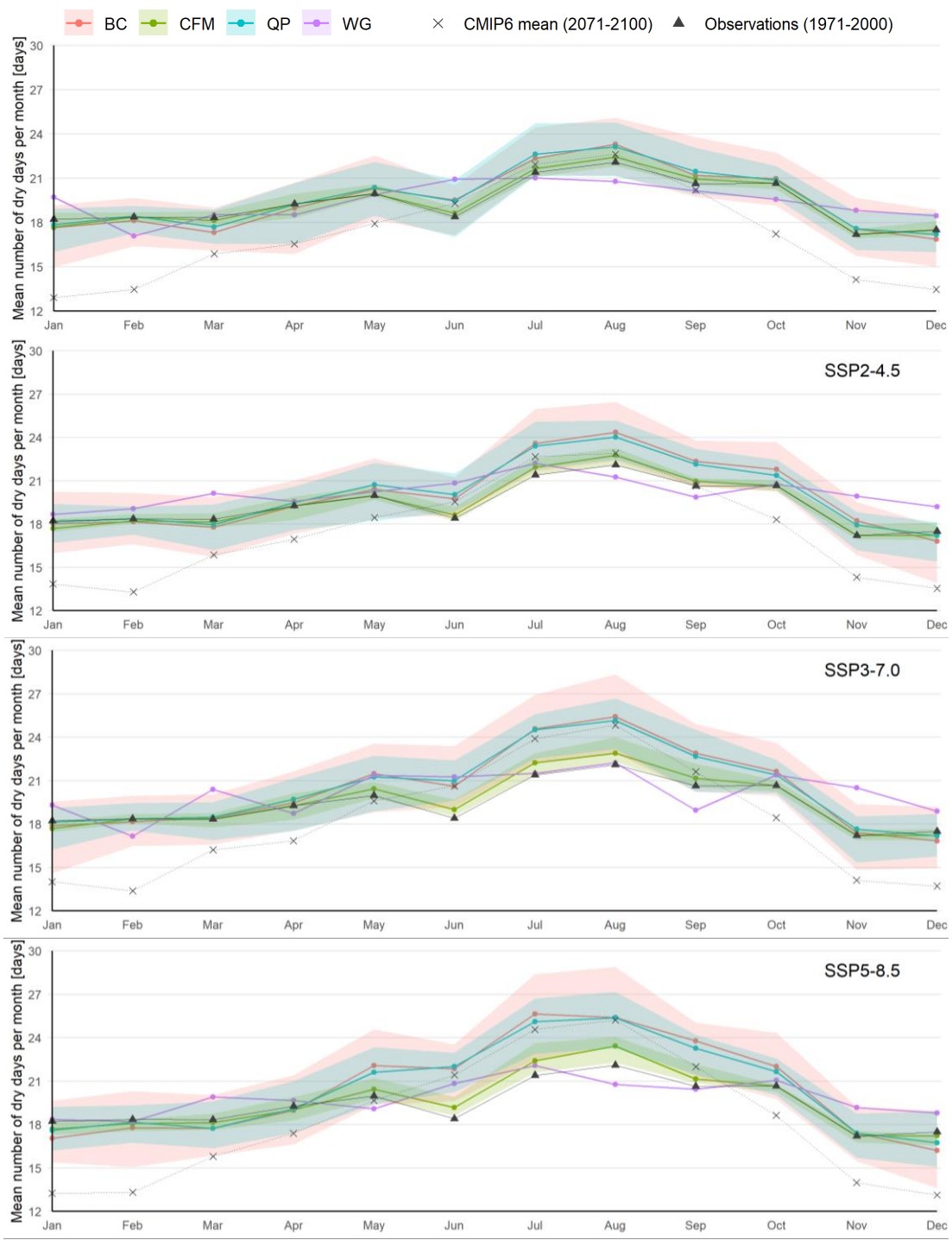

**Figure 5.** Graphical representation of results for number of dry days under different future scenarios. Coloured lines represent median values of the ensemble, shades represent the variation within the ensemble (10% – 90% quantiles). CMIP6 GCM projections (not downscaled; dashed line) and Uccle observations (solid line) are given as reference.

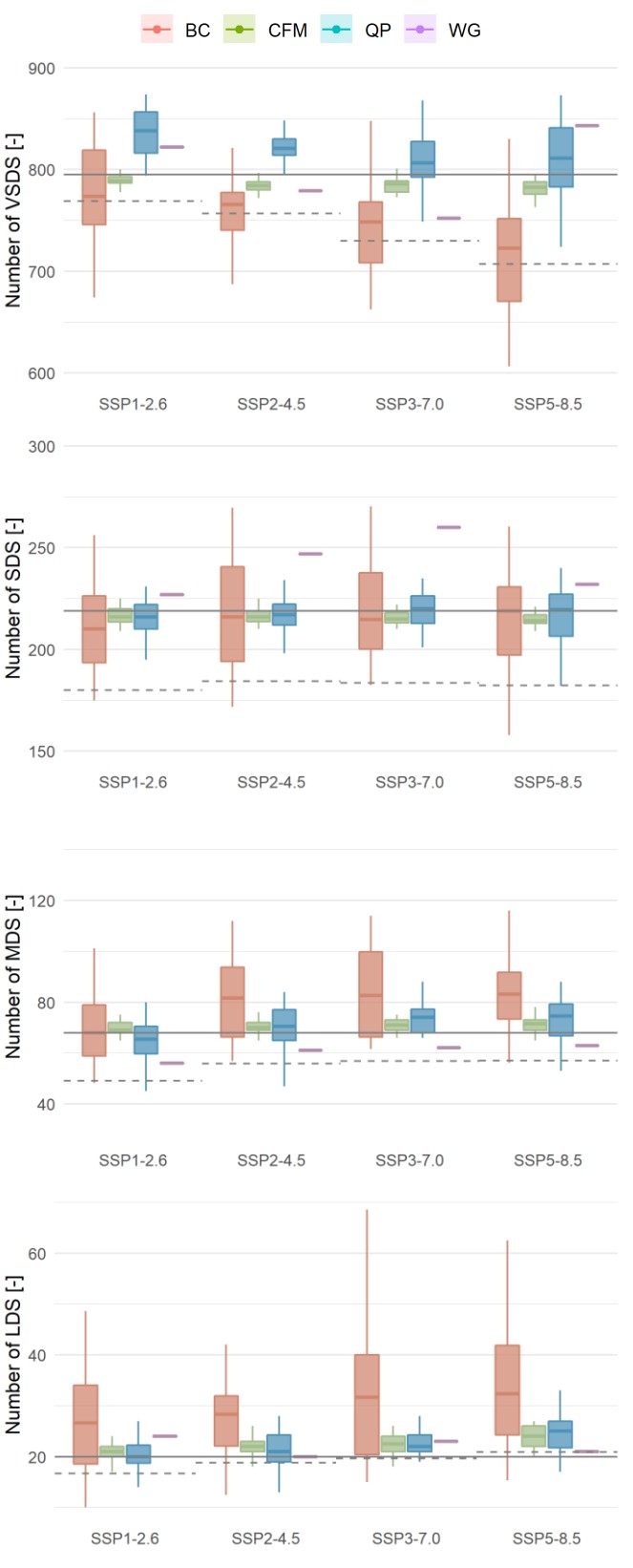

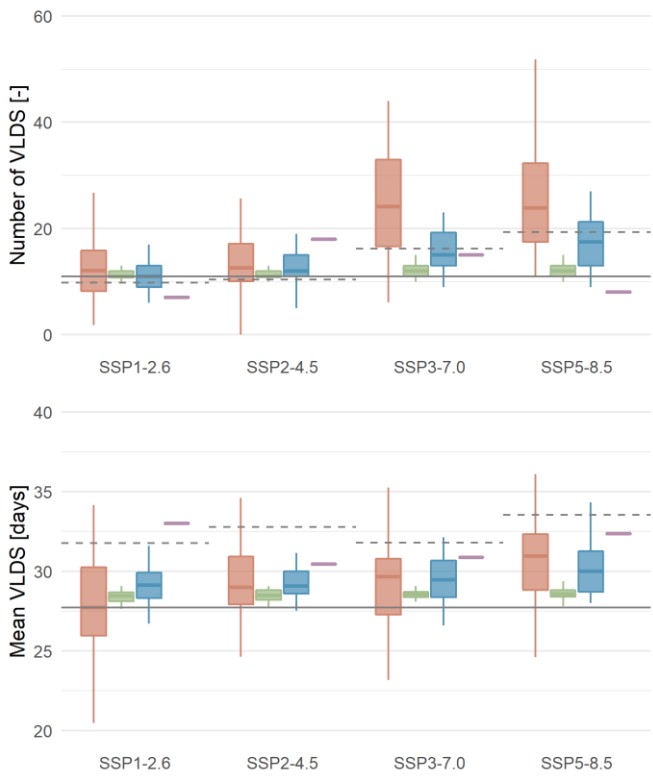

**Figure 6.** Boxplot representation of results for number of dry spells and dry spell length under different future scenarios. WG downscales the ensemble as a whole, resulting in only one data point. CMIP6 GCM projections (not downscaled; dashed line) and Uccle observations (solid line) are given as reference. VSDS, SDS, MDS, LDS and VLDS denote very short, short, medium, long and very long dry spells, respectively. Top and bottom of the box show the 75th and 25th percentiles of the relative error, respectively. Top and bottom of the whiskers show the 5th and 95th percentiles, respectively. Horizontal black line in the middle of the box represents the median.

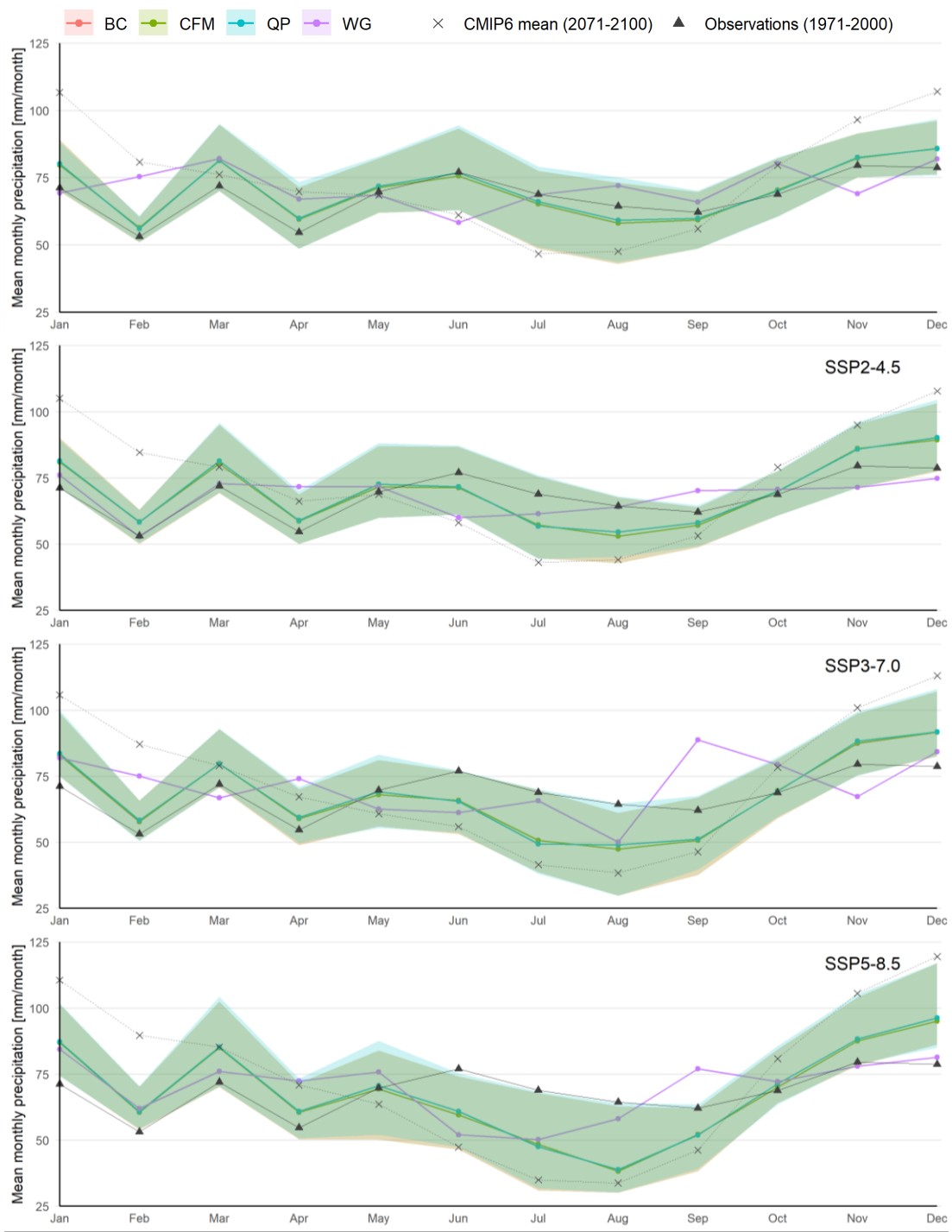

**Figure 7.** Graphical representation of results for total precipitation under different future scenarios. Coloured lines represent median values of the ensemble, shades represent the variation within the ensemble (10% − 90% quantiles). CMIP6 GCM projections (not downscaled; dashed line) and Uccle observations (solid line) are given as reference.

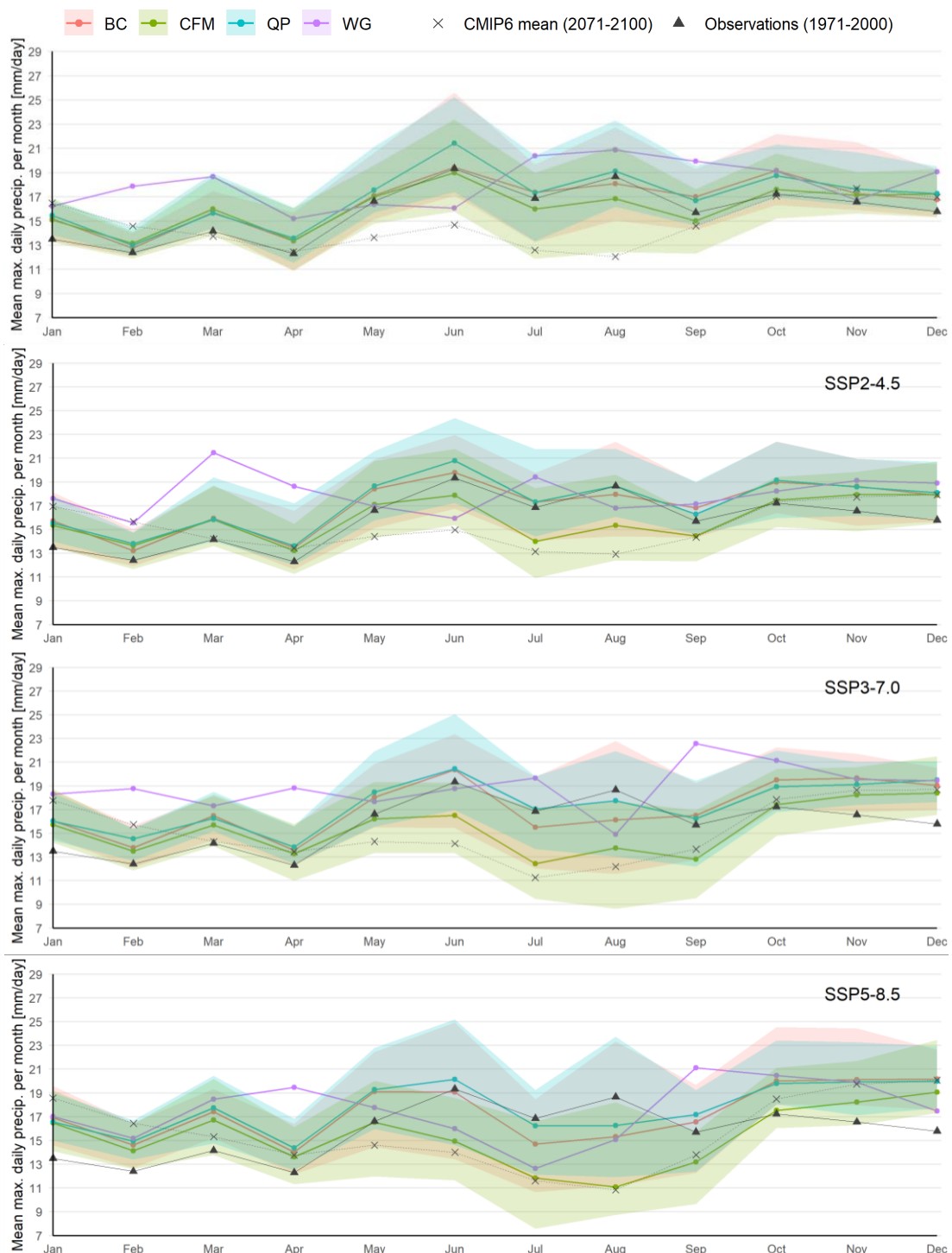

**Figure 8.** Graphical representation of results for maximum daily precipitation under different future scenarios. Coloured lines represent median values of the ensemble, shades represent the variation within the ensemble (10% – 90% quantiles). CMIP6 GCM projections (not downscaled; dashed line) and Uccle observations (solid line) are given as reference.

**Table 1.** Overview of the CMIP6 GCM ensemble used in this study (r: realisation or ensemble member, i: initialisation method, p: physics, f: forcing). The r1i1p1f1 run is used for all the GCMs except five GCMs for which this run is not available and so their r1i1p1f2 and r2i1p1f1 runs are used.

| Model | Resolution | | Variant label | His. | SSP1-2.6 | SSP2-4.5 | SSP3-7.0 | SSP5-8.5 |
|---|---|---|---|---|---|---|---|---|
| | Lat [°] | Lon [°] | | | | | | |
| ACCESS-CM2 | 1.3 | 1.9 | r1i1p1f1 | 1 | 1 | 1 | 1 | 1 |
| ACCESS-ESM1-5 | 1.3 | 1.9 | r1i1p1f1 | 1 | 1 | 1 | 1 | 1 |
| BCC-CSM2-MR | 1.1 | 1.1 | r1i1p1f1 | 1 | 1 | 1 | 1 | 1 |
| CAMS-CSM1-0 | 1.1 | 1.1 | r2i1p1f1 | 1 | 1 | 1 | 1 | 1 |
| CanESM5 | 2.8 | 2.8 | r1i1p1f1 - r25i1p1f1 | 25 | 25 | 25 | 25 | 25 |
| CESM2 | 0.9 | 1.3 | r1i1p1f1 | 1 | 1 | 1 | 1 | 1 |
| CESM2-WACCM | 0.9 | 1.3 | r1i1p1f1 | 1 | 1 | 1 | 1 | 1 |
| CMCC-CM2-SR5 | 1.0 | 1.0 | r1i1p1f1 | 1 | 1 | 1 | 1 | 1 |
| CNRM-CM6-1 | 1.4 | 1.4 | r1i1p1f2 | 1 | 1 | 1 | 1 | 1 |
| CNRM-ESM2-1 | 1.4 | 1.4 | r1i1p1f2 | 1 | 1 | 1 | 1 | 1 |
| EC-Earth3 | 0.7 | 0.7 | r1i1p1f1 | 1 | 1 | 1 | 1 | 1 |
| EC-Earth3-Veg | 0.7 | 0.7 | r1i1p1f1 | 1 | 1 | 1 | 1 | 1 |
| EC-Earth3-Veg-LR | 1.1 | 1.1 | r1i1p1f1 | 1 | 1 | 1 | 1 | 1 |
| FGOALS-g3 | 2.0 | 2.0 | r1i1p1f1 | 1 | 1 | 1 | 1 | 1 |
| GFDL-ESM4 | 1.0 | 1.3 | r1i1p1f1 | 1 | 1 | 1 | 1 | 1 |
| IITM-ESM | 1.9 | 1.9 | r1i1p1f1 | 1 | 1 | 1 | 1 | 1 |
| INM-CM4-8 | 1.5 | 2.0 | r1i1p1f1 | 1 | 1 | 1 | 1 | 1 |
| INM-CM5-0 | 1.5 | 2.0 | r1i1p1f1 | 1 | 1 | 1 | 1 | 1 |
| IPSL-CM6A-LR | 1.3 | 2.5 | r1i1p1f1 | 1 | 1 | 1 | 1 | 1 |
| KACE-1-0-G | 1.3 | 1.9 | r1i1p1f1 | 1 | 1 | 1 | 1 | 1 |
| MIROC6 | 1.4 | 1.4 | r1i1p1f1 | 1 | 1 | 1 | 1 | 1 |
| MIROC-ES2L | 2.8 | 2.8 | r1i1p1f2 | 1 | 1 | 1 | 1 | 1 |
| MPI-ESM1-2-HR | 0.9 | 0.9 | r1i1p1f1 | 1 | 1 | 1 | 1 | 1 |
| MPI-ESM1-2-LR | 1.9 | 1.9 | r1i1p1f1 | 1 | 1 | 1 | 1 | 1 |
| MRI-ESM2-0 | 1.1 | 1.1 | r1i1p1f1 | 1 | 1 | 1 | 1 | 1 |
| NorESM2-LM | 1.9 | 2.5 | r1i1p1f1 | 1 | 1 | 1 | 1 | 1 |
| NorESM2-MM | 0.9 | 1.3 | r1i1p1f1 | 1 | 1 | 1 | 1 | 1 |
| UKESM1-0-LL | 1.9 | 1.3 | r1i1p1f2 | 1 | 1 | 1 | 1 | 1 |

**Table 2.** Target variables used for evaluation of the simulations of WG.

| Target variable | Abbr. | Unit | Weight |
|---|---|---|---|
| Annual number of dry days | and | – | – |
| Seasonal number of dry days, winter | sndwi | – | 0.10 |
| Seasonal number of dry days, spring | sndsp | – | 0.10 |
| Seasonal number of dry days, summer | sndsu | – | 0.20 |
| Seasonal number of dry days, autumn | sndau | – | 0.10 |
| Annual precipitation | ap | mm | – |
| Seasonal precipitation, winter | spwi | mm | 0.03 |
| Seasonal precipitation, spring | spsp | mm | 0.06 |
| Seasonal precipitation, summer | spsu | mm | 0.15 |
| Seasonal precipitation, autumn | spau | mm | 0.06 |
| Annual number of events above 10 mm per day | n10mm | – | 0.10 |
| Annual number of events above 20 mm per day | n20mm | – | 0.05 |
| Annual maximum daily precipitation | mdp | mm | 0.05 |

**Table 3.** Overview of the considered research indicators.

| Research indicators | # |
|---|---|
| Mean monthly number of dry days | 12 |
| Number of dry spells per class | 5 |
| Mean length of very long dry spells | 1 |
| Mean monthly precipitation | 12 |
| Maximum monthly precipitation | 12 |
| | 42 |

**Table 4.** Classification of dry spells based on their length along with the limits for each class derived from observed time series.

| Class name | Percentiles | Limits [days] |
|---|---|---|
| Very short dry spell | < 20th | [2, 7] |
| Short dry spell | 20th – 40th | [8, 13] |
| Medium dry spell | 40th – 60th | [14, 19] |
| Long dry spell | 60th – 80th | [20, 25] |
| Very long dry spell | > 80th | [26, ∞] |

**Table 5.** Climate change signals and corresponding significance for BC and CFM. Climate change signal is the change relative to the historical observations (1971-2000). Numbers in italic, bold and bold italic denote significant changes at 20%, 10% and 5% levels, respectively.

| Research indicator | | Obs. | BC | | | | CFM | | | |
|---|---|---|---|---|---|---|---|---|---|---|
| | | | SSP1-2.6 | SSP2-4.5 | SSP3-7.0 | SSP5-8.5 | SSP1-2.6 | SSP2-4.5 | SSP3-7.0 | SSP5-8.5 |
| Dry spell length | | 27 | 4.7% | 7.5% | 4.9% | 10.5% | *2.3%* | *2.8%* | *2.8%* | **3.2%** |
| Number of dry spells | Very short | 795 | *6.8%* | 5.3% | 1.9% | -1.6% | -0.6% | **-1.3%** | *-1.3%* | *-1.6%* |
| | Short | 219 | 6.2% | 8.0% | 8.1% | 7.6% | -1.1% | -1.2% | -1.7% | **-2.3%** |
| | Medium | 68 | *27.9%* | **47.9%** | **48.5%** | **50.7%** | 3.2% | 3.3% | *4.4%* | *5.2%* |
| | Long | 20 | ***60.9%*** | **77.5%** | **90.1%** | **86.6%** | 5.0% | **10.5%** | **13.0%** | ***20.2%*** |
| | Very long | 11 | 6.8% | 5.3% | 1.9% | -1.6% | 1.6% | 3.9% | *10.7%* | 10.4% |
| Number of dry days | Jan | 18 | -4.6% | -0.9% | -2.5% | -5.5% | -1.4% | -1.6% | *-2.3%* | ***-3.0%*** |
| | Feb | 18 | -1.3% | -0.6% | -0.5% | -2.4% | 0.1% | -0.5% | -0.5% | -1.8% |
| | Mar | 18 | -4.7% | -3.2% | -1.2% | -3.0% | -0.6% | -0.6% | -0.6% | -1.2% |
| | Apr | 19 | -2.6% | -1.3% | 0.9% | -0.9% | -0.5% | -0.5% | -0.1% | -1.0% |
| | May | 20 | 1.4% | 2.6% | *7.1%* | **9.5%** | 0.9% | 0.9% | 1.7% | 2.1% |
| | Jun | 18 | 4.5% | **8.2%** | ***13.3%*** | ***18.0%*** | 1.1% | *1.9%* | *3.4%* | **4.9%** |
| | Jul | 21 | 5.9% | **10.3%** | **15.0%** | **19.3%** | 1.5% | **2.5%** | **3.8%** | **5.3%** |
| | Aug | 22 | 5.2% | 9.6% | ***14.9%*** | **17.5%** | 1.5% | 2.7% | ***4.2%*** | **5.5%** |
| | Sep | 21 | 4.1% | 7.6% | *10.3%* | **13.6%** | 1.2% | *1.8%* | **3.0%** | **3.6%** |
| | Oct | 21 | 1.4% | 5.7% | 5.2% | 6.2% | -0.2% | 0.2% | -0.1% | -0.2% |
| | Nov | 17 | 2.5% | 3.2% | 1.3% | 1.5% | 0.6% | 0.5% | -0.1% | -0.5% |
| | Dec | 18 | -3.6% | -6.1% | -3.7% | -6.1% | -0.3% | -0.8% | -1.2% | -2.1% |
| Total precipitation | Jan | 71.2 | 11.4% | 12.9% | **17.8%** | ***23.1%*** | 11.4% | 12.9% | **17.8%** | ***23.1%*** |
| | Feb | 53.1 | 5.2% | 7.5% | 9.8% | 17.1% | 5.3% | 7.5% | 9.8% | 17.2% |
| | Mar | 72 | **14.0%** | 12.6% | 13.3% | *20.0%* | **13.9%** | 12.6% | 13.3% | 20.0% |
| | Apr | 54.7 | 9.6% | 9.7% | 8.6% | 12.3% | 9.6% | 9.7% | 8.6% | 12.2% |
| | May | 69.7 | 3.1% | 4.6% | -1.6% | -2.6% | 3.1% | 4.6% | -1.6% | -2.6% |
| | Jun | 77.1 | 0.7% | -5.9% | -14.0% | -22.2% | 0.7% | -5.9% | -14.0% | -22.2% |
| | Jul | 68.9 | -7.3% | *-14.0%* | ***-23.2%*** | ***-31.7%*** | -7.3% | *-14.0%* | ***-23.2%*** | ***-31.7%*** |
| | Aug | 64.4 | -8.5% | -16.5% | ***-27.1%*** | ***-32.8%*** | -8.5% | -16.5% | ***-27.2%*** | ***-32.8%*** |
| | Sep | 62.1 | -3.1% | -8.4% | *-15.6%* | *-19.3%* | -3.1% | -8.4% | *-15.6%* | **-19.3%** |
| | Oct | 68.8 | 4.4% | 0.7% | 2.3% | 5.0% | 4.4% | 0.7% | 2.3% | 5.0% |
| | Nov | 79.6 | 4.2% | 6.1% | 10.2% | 13.4% | 4.2% | 6.1% | 10.2% | 13.4% |
| | Dec | 78.7 | *9.8%* | *14.2%* | **17.9%** | ***24.5%*** | *9.8%* | *14.2%* | **17.9%** | ***24.5%*** |
| Maximum daily precipitation | Jan | 13.5 | *12.0%* | **17.4%** | ***21.5%*** | ***25.3%*** | 11.4% | 12.9% | **17.8%** | ***23.1%*** |
| | Feb | 12.4 | 4.0% | 7.6% | 11.3% | 17.7% | 5.3% | 7.5% | 9.8% | 17.2% |
| | Mar | 14.2 | *11.6%* | 13.9% | *15.8%* | *22.5%* | **13.9%** | 12.6% | 13.3% | *20.0%* |
| | Apr | 12.3 | 8.4% | 12.2% | 12.1% | 15.3% | 9.6% | 9.7% | 8.6% | 12.2% |
| | May | 16.6 | 5.3% | 9.6% | 8.3% | 12.9% | 3.1% | 4.6% | -1.6% | -2.6% |
| | Jun | 19.3 | 8.1% | 4.5% | 2.1% | -1.2% | 0.7% | -5.9% | -14.0% | -22.2% |
| | Jul | 16.9 | 0.5% | 0.9% | -4.4% | -12.5% | -7.3% | *-14.0%* | ***-23.2%*** | ***-31.7%*** |
| | Aug | 18.7 | 0.0% | -3.8% | -9.4% | -12.9% | -8.5% | -16.5% | ***-27.1%*** | ***-32.8%*** |
| | Sep | 15.7 | 6.4% | 6.5% | 3.1% | 4.0% | -3.1% | -8.4% | *-15.6%* | **-19.3%** |
| | Oct | 17.2 | 11.2% | 10.7% | 14.4% | 20.3% | 4.4% | 0.7% | 2.3% | 5.0% |
| | Nov | 16.6 | 8.9% | 11.2% | 17.0% | *23.7%* | 4.2% | 6.1% | 10.2% | 13.4% |
| | Dec | 15.8 | 8.2% | 13.1% | **19.7%** | **26.6%** | *9.8%* | *14.2%* | **17.9%** | *24.5%* |

**Table 6**. Climate change signals for QP and WG and corresponding significance for QP. Significance testing is not possible for WG as it does not downscale each member of the ensemble separately. Climate change signal is the change relative to the historical observations (1971-2000). Numbers in italic, bold and bold italic denote significant changes at 20%, 10% and 5% levels, respectively.

| Research indicator | | Obs. | QP | | | | WG | | | |
|---|---|---|---|---|---|---|---|---|---|---|
| | | | SSP1-2.6 | SSP2-4.5 | SSP3-7.0 | SSP5-8.5 | SSP1-2.6 | SSP2-4.5 | SSP3-7.0 | SSP5-8.5 |
| Dry spell length | | 27 | 4.9% | **5.6%** | *6.5%* | **8.7%** | 19.0% | 9.8% | 11.3% | 16.8% |
| Number of dry spells | Very short | 795 | **5.5%** | *4.8%* | 2.0% | 1.4% | 3.4% | -2.0% | -5.4% | 6.0% |
| | Short | 219 | -1.6% | -1.4% | 0.0% | -1.2% | 3.7% | 12.8% | 18.7% | 5.9% |
| | Medium | 68 | -4.3% | 2.5% | 6.5% | 6.6% | -17.6% | -10.3% | -8.8% | -7.4% |
| | Long | 20 | 3.6% | 5.4% | 16.1% | 21.4% | 20.0% | 0.0% | 15.0% | 5.0% |
| | Very long | 11 | 3.9% | 18.5% | *43.8%* | **62.7%** | -36.4% | 63.6% | 36.4% | -27.3% |
| Number of dry days | Jan | 18 | -2.6% | -0.7% | -1.8% | -3.4% | 8.2% | 2.4% | 5.9% | 0.7% |
| | Feb | 18 | -0.2% | 0.1% | 0.1% | -1.0% | -6.9% | 3.8% | -6.5% | -0.9% |
| | Mar | 18 | -3.6% | -2.9% | -1.2% | -2.5% | 1.1% | 9.8% | 11.3% | 8.5% |
| | Apr | 19 | -1.6% | -0.5% | 0.7% | -0.3% | -3.8% | 1.6% | -2.8% | 2.1% |
| | May | 20 | 1.8% | 2.4% | 6.1% | 7.6% | -0.3% | 1.3% | 7.0% | -4.3% |
| | Jun | 18 | 4.1% | *8.0%* | *12.9%* | **18.0%** | 13.8% | 13.2% | 15.6% | 13.2% |
| | Jul | 21 | 5.7% | *9.0%* | *13.1%* | **16.6%** | -1.7% | 3.7% | 0.5% | 3.1% |
| | Aug | 22 | *4.8%* | **8.1%** | *12.8%* | *14.7%* | -5.9% | -3.8% | 0.6% | -6.0% |
| | Sep | 21 | *4.3%* | **6.9%** | *9.5%* | *11.8%* | -2.3% | -3.7% | -8.1% | -1.0% |
| | Oct | 21 | 0.7% | 3.6% | 3.1% | 4.0% | -5.3% | 0.5% | 3.5% | 1.9% |
| | Nov | 17 | 1.9% | 2.2% | 0.6% | 0.7% | 9.5% | 15.9% | 19.2% | 11.4% |
| | Dec | 18 | -1.1% | -2.7% | -1.8% | -3.4% | 5.5% | 9.7% | 8.0% | 7.4% |
| Total precipitation | Jan | 71.2 | 11.5% | 13.3% | **18.1%** | **23.3%** | -2.6% | 6.8% | 15.1% | 18.5% |
| | Feb | 53.1 | 5.5% | 7.8% | 10.4% | 17.7% | 42.0% | -0.7% | 41.3% | 16.8% |
| | Mar | 72 | **14.3%** | 13.1% | 13.8% | *21.1%* | 14.1% | 1.2% | -7.1% | 5.6% |
| | Apr | 54.7 | 10.4% | 10.8% | 9.7% | 13.5% | 22.6% | 31.1% | 35.5% | 32.4% |
| | May | 69.7 | 4.6% | 5.9% | 0.1% | -0.8% | -1.8% | 3.0% | -10.3% | 8.7% |
| | Jun | 77.1 | 1.6% | -5.1% | -13.4% | *-21.1%* | -24.3% | -22.1% | -20.5% | -32.6% |
| | Jul | 68.9 | -6.6% | -13.7% | **-23.3%** | **-31.3%** | -0.2% | -10.7% | -4.5% | -27.0% |
| | Aug | 64.4 | -7.0% | -14.7% | **-25.5%** | **-32.2%** | 11.9% | -0.6% | -22.3% | -9.7% |
| | Sep | 62.1 | -2.4% | -7.3% | -14.4% | **-18.0%** | 6.2% | 13.0% | 42.9% | 24.0% |
| | Oct | 68.8 | 5.1% | 1.5% | 2.8% | 6.1% | 16.6% | 2.8% | 15.4% | 4.9% |
| | Nov | 79.6 | 4.8% | 6.5% | 10.6% | 14.0% | -13.2% | -10.2% | -15.4% | -1.9% |
| | Dec | 78.7 | *10.1%* | *14.6%* | **18.5%** | **25.3%** | 4.1% | -4.9% | 7.1% | 3.3% |
| Maximum daily precipitation | Jan | 13.5 | *13.5%* | **17.0%** | **21.3%** | **26.2%** | 20.2% | 30.4% | 35.6% | 26.1% |
| | Feb | 12.4 | 6.2% | 9.9% | *13.7%* | *20.9%* | 44.1% | 25.6% | 51.4% | 22.4% |
| | Mar | 14.2 | **14.4%** | 14.8% | *18.1%* | **25.3%** | 31.8% | 51.5% | 22.3% | 30.5% |
| | Apr | 12.3 | 11.9% | 14.2% | 14.5% | *17.5%* | 23.6% | 51.5% | 52.9% | 58.2% |
| | May | 16.6 | 8.1% | 12.8% | 12.8% | 16.2% | -1.6% | 2.1% | 6.3% | 7.0% |
| | Jun | 19.3 | 11.8% | 8.3% | 6.8% | 2.8% | -16.9% | -17.5% | -2.8% | -17.2% |
| | Jul | 16.9 | 3.5% | 4.6% | -1.4% | -5.2% | 20.9% | 15.2% | 16.6% | -25.0% |
| | Aug | 18.7 | 4.6% | 0.9% | -2.6% | -9.4% | 11.9% | -9.9% | -20.1% | -19.3% |
| | Sep | 15.7 | 6.8% | 6.8% | 1.6% | 3.5% | 26.9% | 9.2% | 43.6% | 34.5% |
| | Oct | 17.2 | 10.6% | 10.8% | 12.7% | 21.4% | 11.0% | 5.8% | 22.8% | 18.8% |
| | Nov | 16.6 | 10.5% | 11.8% | *16.8%* | *21.6%* | 1.1% | 15.4% | 18.0% | 20.1% |
| | Dec | 15.8 | *10.8%* | 14.9% | **22.2%** | **28.9%** | 20.8% | 19.8% | 23.2% | 10.7% |

**Table 7.** Maximum deviation relative to the mean change factor projected by the CMIP6 ensemble allowed for acceptance for each target variable in WG. These deviations correspond to a 95% confidence interval of the distribution of each target variable projection within the GCM ensemble.

| Target variable | | Abbr. | Weight | SSP1-2.6 | SSP2-4.5 | SSP3-7.0 | SSP5-8.5 |
|---|---|---|---|---|---|---|---|
| Dry days | Annual | and | 0 | 3.43% | 3.22% | 4.15% | 4.40% |
| | Winter | sndwi | 0.1 | 5.21% | 6.89% | 7.18% | 8.15% |
| | Spring | sndsp | 0.1 | 6.33% | 4.85% | 5.78% | 7.15% |
| | Summer | sndsu | 0.2 | 5.86% | 6.05% | 8.14% | 8.24% |
| | Autumn | sndau | 0.1 | 4.62% | 3.75% | 5.75% | 5.58% |
| Total | Annual | and | 0 | 4.35% | 5.07% | 6.24% | 6.46% |
| precipitation | Winter | sndwi | 0.03 | 6.60% | 7.72% | 8.61% | 10.64% |
| | Spring | sndsp | 0.06 | 10.00% | 9.54% | 10.44% | 12.38% |
| | Summer | sndsu | 0.15 | 11.95% | 10.98% | 14.68% | 14.87% |
| | Autumn | sndau | 0.06 | 7.20% | 6.82% | 7.47% | 8.21% |
| Extreme | 10 mm | n10mm | 0.1 | 8.49% | 12.08% | 13.62% | 15.01% |
| precipitation | 20 mm | n20mm | 0.05 | 27.59% | 35.70% | 36.61% | 48.14% |
| | Max | mdp | 0.05 | 7.42% | 11.32% | 11.39% | 13.29% |