# Peer review of "Comparison of statistical downscaling methods for climate change impact analysis on precipitation-driven drought"

_Hydrology and Earth System Sciences, 2020_

## Referee Comment (RC1) · Anonymous Referee #1 · 19 Nov 2020

HESS-2020-506 Title: Comparison of statistical downscaling methods for climate change impact analysis on drought Authors: Hossein Tabari, Daan Buekenhout, Patrick Willems

GENERAL COMMENT The paper presents a comparative analysis of four (4) Statistical Downscaling Methods (SDMs), namely, Bias Correction (BC), Change Factor of Mean (CFM), Quantile Perturbation (QP) and Event Based Weather Generator (EBWG) to assess climate change impact on drought by the end of the 21st century (2071-2100) relative to a baseline period of 1971-2000. The SDMs were applied to downscale daily precipitation from 14-member ensemble of CMIP6 GCM at the Uccle weather station in Belgium for four future scenarios, namely, SSP1-2.6, SSP2-4.5, SSP3-7.0 and SSP5-8.5. Various drought indices have been calculated and used in the

comparison of SDMs' results for the future period with the drought indices estimated from the observed precipitation for the baseline period.

The paper is well organized, written and comprehensive. However, there are a few points that should be clarified and addressed. Overall, the paper merits publication in the HESS after the moderate comments are properly addressed.

SPECIFIC COMMENTS 1) Title: The title of the paper should include the word "precipitation". Precipitation is the variable downscaled from GCMs in the paper. Additionally, drought phenomenon is affected by other meteorological variables, for example temperature, evapotranspiration and others. It should be clear from the title that the work presented in the paper deals with the downscaling of precipitation for the estimation of climate change impacts on droughts. 2) Abstract: It should be made clear in the Abstract that the downscaling exercise was made for the weather station of Uccle located in Belgium. This information is missing from the stand-alone abstract of the paper. 3) It is usual to calibrate/validate the SDMs during the historical base period (1971-2000 in the paper) and, then, apply them for the future period(s). The authors although mentioned that the methods have been calibrated using the observed precipitation at the Uccle weather station, they do not present any results (i.e. statistics, graphs) about the calibration of the SDMs. The presentation of the calibration results are necessary to assess the validity of the downscaling methods before using them to the future periods.

MINOR COMMENTS 4) Lines 46-48. It is written "Precipitation and the number of wet days were found to increase during summer and to decrease during winter, while evapotranspiration was found to increase for both seasons. This suggests drier summers and wetter winters." This statement is quite vague and needs further explanation on the ratio of precipitation and evapotranspiration to generate drier summers and wetter winters. 5) Line 130. The equation of $\alpha m$ should be written better. The equal sign is not shown properly. 6) Table 3. What is WLDS? Although the term is presented in the text of the paper, it should be written in full. Tables are stand-alone elements of a paper. 7) Figure 2. The color bars and the reference lines should be explained

in the figure legend and/or the figure caption. 8) Tables 5 and 6. Please indicate the significance levels at which the changes have been tested, otherwise the information conveyed by these tables is vague. You may put indicators, a, b, c, and a note that a is for 5% significance level, b-10% s.l. and c-20% s.l.

―――――――――――――――――――

---

## Referee Comment (RC2) · Anonymous Referee #2 · 11 Dec 2020

In general, I found this manuscript difficult to follow. If I weren't supposed to review it, I would have stopped reading it fairly quickly. This is not so much a question about the English but the way the material is structured and that it presumes that the reader already is familiar with the work. I also think I have some misgivings about the results and have a few issues with the nomenclature. There is no proper evaluation of the methods, and I think the authors could give more background and discuss their efforts in the context of other relevant papers.

L.40-48: It may be of interest to note that the occurrence of droughts also may be a consequence of a reduction in the global area with 24-hr rainfall (DOI: 10.1088/1748-9326/aab375). L.66: The statement 'As there is no single best downscaling method' may be correct generally for a range of situations, but for specific and limited problems,

there may be some method that is superior and provides the most reliable answer. L.73/79: ensemble sizes of 14 or 25 members are on the low end for giving robust information, due to pronounced and stochastic/chaotic decadal variations (Deser et al., 2012) and 'the law of small numbers'. If the ensemble is to be regarded as a statistical sample, then it typically needs more than 30 data points to get results that are not heavily affected by random sampling fluctuation, and preferably more than 100 members to get more robust statistics. L83: If 'precipitation time series produced by GCMs as sole predictor' also is used for model calibration, then there seems to be a problem because the GCMs are not synchronised with the observations in the real world. L92: Both GCMs and RCMs have a minimum skillful scale (DOI: 10.2151/jmsj.2015-042) which implies that single grid box values should not be used as a predictor. The difference between downscaling and bias adjustment is that the former utilises the scales that the models are able to skillfully reproduce and known dependencies between large and small scales to infer changes in the local (small-scale) climate whereas the latter is about modifying individual grid-box values (below the minimum skillful scales) to match the statistical properties of observations from similar locations. L100: I found the section difficult to follow and at times there seemed to be too little information. A rule of thumb is it is annoying to read a paper where you need to read another paper before you understand what it does, and the reader should be able to reproduce the results based on a detailed 'recipe' explaining the method. Reproducibility is an important issue. L101: State that the four methods are explained in more detail below. Also, I'm not sure why there are four methods - what is the merit of each? Sure, I get it that they may give different results, but so what? L107: Insufficient information about the BC methods? L115: A change factor of mean cannot really be classified as a downscaling method as it does not involve different spatial scales. L180: Perhaps an illustration will make it easier to follow. L199: Why not downscale the parameters of the pdf for the dry-spell duration? If there is a probability p that it rains on a specific day, then the pdf for the duration of dry spells ought to follow a binomial distribution with one parameter p = 1/mean(spell lengths). This approach has been tried for heatwaves in India (DOI:

10.5194/ascmo-4-37-2018). Because the statistics of spells (duration of events) tend to follow the binomial distribution, it is advisable to include the mean duration - not just its median (which is usually not a quantity that corresponds to the parameters of a pdf). L200-206: There is a subtle but important difference between a rain gauge measurement (a point measurement) and gridbox values from GCMs (area mean estimates), which also has implications for threshold values. This is also expected to be affected by different spatial resolutions (Table 1). For a coarser model, local 'convective' would be smudged out over a larger area (I guess it is parameterised) and expected to have a different statistical characteristic to a very tiny spatial sample (rain gauge measurements with diameters of the order of centimetres). This in itself is a justification for bias adjustment, but nevertheless makes it difficult to compare dry and wet days in models with different spatial resolution. L222: Pmax is the max monthly maximum precipitation or the mean monthly maximum? L235: Why is a two-tailed test used here? L237: It is important with a proper evaluation of the methods involving e.g. cross-validation and historical data, which seems to be missing. E.g., can the methods reproduce historical changes/variations in dry-spell lengths? L253: 'medium' should be 'median'? It's better to use the mean because the statistics of dry-spell length is expected to follow a distribution that is not too different to the binomial distribution for which the mean is connected to its only parameter. L258: Define 'DSL'. L269: It's annoying to stumble across acronyms like 'RI' which I then need to scroll up to remind me of what it stands for. The same goes with the other acronyms. I think it's a bad habit (and sloppy) to use many acronyms and abbreviations and if there is no need to do so, it's generally better to spell out the words to make it easier for the reader and ensure that (s)he focuses on the message rather than trying to decipher the text. The punishment is of course revisions... I think that the ensemble used here is too small to make any judgement about future climate due to pronounced stochastic local variability on decadal scales (Deser et al., 2012) and the law of small numbers. This stochastic nature should call for the use of probabilistic projections. Furthermore, I would not call the efforts discussed here 'downscaling' but they are more along the line of 'prediction' or 'bias correction' in

my opinion (there is no scaling dependencies being utilised between large and small). It can be shown that the large-scale circulation (SLP anomalies) tend to determine whether there is rain or no rain for a location, and downscaling would involve using the large-scale circulation as a means to infer the wet-day frequency and maybe also dry-spell lengths. L380: Need to define 'accurately' and provide evidence for the statement.

---

## Author Comment (AC1) · 30 Dec 2020

We thank the reviewer for the detailed and constructive review. In the following, we express our view on the raised points, and illustrate our plan on how to improve the paper.

GENERAL COMMENT

The paper presents a comparative analysis of four (4) Statistical Downscaling Methods (SDMs), namely, Bias Correction (BC), Change Factor of Mean (CFM), Quantile Perturbation (QP) and Event Based Weather Generator (EBWG) to assess climate change impact on drought by the end of the 21st century (2071-2100) relative to a baseline period of 1971-2000. The SDMs were applied to downscale daily precipita-

tion from 14-member ensemble of CMIP6 GCM at the Uccle weather station in Belgium for four future scenarios, namely, SSP1-2.6, SSP2-4.5, SSP3-7.0 and SSP5-8.5. Various drought indices have been calculated and used in the comparison of SDMs' results for the future period with the drought indices estimated from the observed precipitation for the baseline period. The paper is well organized, written and comprehensive. However, there are a few points that should be clarified and addressed. Overall, the paper merits publication in the HESS after the moderate comments are properly addressed.

Reply: We thank the reviewer for the encouraging assessment and helpful comments.

SPECIFIC COMMENTS

1) Title: The title of the paper should include the word "precipitation". Precipitation is the variable downscaled from GCMs in the paper. Additionally, drought phenomenon is affected by other meteorological variables, for example temperature, evapotranspiration and others. It should be clear from the title that the work presented in the paper deals with the downscaling of precipitation for the estimation of climate change impacts on droughts.

Reply: We agree with the reviewer that the title of the paper should show that precipitation and its lack are investigated in this study. The title will be revised accordingly.

2) Abstract: It should be made clear in the Abstract that the downscaling exercise was made for the weather station of Uccle located in Belgium. This information is missing from the stand-alone abstract of the paper.

Reply: The weather station information will be added to the abstract.

3) It is usual to calibrate/validate the SDMs during the historical base period (1971-2000 in the paper) and, then, apply them for the future period(s). The authors although mentioned that the methods have been calibrated using the observed precipitation at the Uccle weather station, they do not present any results (i.e. statistics, graphs) about the calibration of the SDMs. The presentation of the calibration results are necessary to

assess the validity of the downscaling methods before using them to the future periods.

Reply: We agree that assessing the validity of the SDMs is important. In the next version of the manuscript, we will present the calibration results of the SDMs.

MINOR COMMENTS

4) Lines 46-48. It is written "Precipitation and the number of wet days were found to increase during summer and to decrease during winter, while evapotranspiration was found to increase for both seasons. This suggests drier summers and wetter winters." This statement is quite vague and needs further explanation on the ratio of precipitation and evapotranspiration to generate drier summers and wetter winters.

Reply: The statement will be clarified by adding more details on the precipitation and evapotranspiration changes.

5) Line 130. The equation of m should be written better. The equal sign is not shown properly.

Reply: A space will be added before the equal sign.

6) Table 3. What is WLDS? Although the term is presented in the text of the paper, it should be written in full. Tables are stand-alone elements of a paper.

Reply: VLDS refers to very long dry spell. It will be written in full in the table. The abbreviations in the other tables and figures will be either written in full or (wherever it is not possible due to a space limitation) defined in the future caption and table title.

7) Figure 2. The color bars and the reference lines should be explained

Reply: Sorry for the missing legend. The legend will be added to the figure.

---

## Author Comment (AC2) · 30 Dec 2020

We thank the reviewer for the detailed and constructive review. In the following, we express our view on the raised points, and illustrate our plan on how to improve the paper.

COMMENTS:

In general, I found this manuscript difficult to follow. If I weren't supposed to review it, I would have stopped reading it fairly quickly. This is not so much a question about the English but the way the material is structured and that it presumes that the reader already is familiar with the work. I also think I have some misgivings about the results and have a few issues with the nomenclature. There is no proper evaluation of the

methods, and I think the authors could give more background and discuss their efforts in the context of other relevant papers.

Rely: We thank the reviewer for the critical assessment and constructive feedback. The major improvements planned for the next version of the manuscript are:

- We will enlarge the ensemble sample size to limit the effect of the random sampling fluctuation, making the total ensemble sample size equal to 53 for each RCP-SSP scenario (simulations from a 29-member ensemble of the CMIP6 GCMs plus 24 extra simulations from the CanESM5 GCM ensemble). In this regard, 'the law of small numbers' and the risk of using a limited sample size will be noted by citing the relevant references.

- We will explain the four methods in more details and justify their choice. In this regard, sufficient detail will be added to enable readers to fully understand the work and to permit the work be replicated by suitably skilled researchers.

- The calibration results of the SDMs will be presented to show how the SDMs can reproduce historical dry day frequency, dry spell duration and total precipitation.

- To improve the readability of the manuscript, acronyms and abbreviations will be written in the full term as much as possible and acronyms and abbreviations will be used only when absolutely necessary (e.g., space limitation on plots).

- The results of the study will be discussed in the context of other relevant studies and the literature review will be modified.

L.40-48: It may be of interest to note that the occurrence of droughts also may be a consequence of a reduction in the global area with 24-hr rainfall (DOI: 10.1088/1748-9326/aab375).

Rely: We appreciate the reviewer for bringing this point to our attention. The drought as a result of a decrease in daily precipitation area will be noted in the next version of the paper.

L.66: The statement 'As there is no single best downscaling method' may be correct generally for a range of situations, but for specific and limited problems, there may be some method that is superior and provides the most reliable answer.

Rely: The sentence refers to the lack of a single best downscaling method for all applications and regions. We agree with the reviewer that some methods are superior for specific applications. That is why we need to evaluate different statistical downscaling methods to select the optimal one for each application based on the information needs (e.g., desired spatial and temporal resolutions) and on available resources (data, expertise, computing resources and time-frames). The sentence will be reformulated in the next version of the paper to clarify this point.

L.73/79: ensemble sizes of 14 or 25 members are on the low end for giving robust information, due to pronounced and stochastic/chaotic decadal variations (Deser et al., 2012) and 'the law of small numbers'. If the ensemble is to be regarded as a statistical sample, then it typically needs more than 30 data points to get results that are not heavily affected by random sampling fluctuation, and preferably more than 100 members to get more robust statistics.

Rely: We agree with the reviewer that small samples subject to "the law of small numbers" are susceptible to the presence of strong random statistical fluctuations and can provide misleading results (Benestad et al., 2017a, b). Our climate model ensemble includes 38 runs from 15 GCM for each of the four considered RCP-SSP scenarios (SSP1-2.6, SSP2-4.5, SSP3-7.0 and SSP5-8.5). Checking the ESGF website shows that 14 additional CMIP6 GCMs are available, which will be added to the previous ensemble. This makes the total ensemble sample size equal to 53 for each RCP-SSP scenario (simulations from a 29-member ensemble of the CMIP6 GCMs plus 24 extra simulations from the CanESM5 GCM ensemble), which is large enough to limit the effect of the random sampling fluctuation. The law of small numbers and the risk of using a limited sample size will also be noted by citing the relevant references.

L83: If 'precipitation time series produced by GCMs as sole predictor' also is used for model calibration, then there seems to be a problem because the GCMs are not synchronised with the observations in the real world.

Rely: Rummukainen (1997) classified SDMs into two groups of perfect prognosis (PP) and model output statistics (MOS) based on the information used for downscaling, and Wilby and Wigely (1997) and Fowler et al. (2007) classified them into three groups of regression methods, weather type approaches, and stochastic weather generators (WGs) based on the relationship used to connect large and local scales. Maraun et al. (2010) combined these two types of the SDM classification and came up with three categories of PP, MOS, and WGs. According to the last classification, the SDMs 1-3 in this study are classified as MOS methods and the SDM 4 as a WG.

As mentioned in paragraph 63 in Maraun et al. (2010), "Depending on the type of simulations used for MOS calibration the predictors can either be simulated precipitation time series or properties of the simulated intensity distribution. Similarly, predictands can either be local precipitation series or properties of the local-scale intensity distribution."

In WGs, the change factors are used to modify observations for a future climate. "Once these change factors are calculated, no large-scale drivers are needed to generate weather time series." (paragraph 81 in Maraun et al., 2010).

L92: Both GCMs and RCMs have a minimum skillful scale (DOI: 10.2151/jmsj.2015-042) which implies that single grid box values should not be used as a predictor. The difference between downscaling and bias adjustment is that the former utilises the scales that the models are able to skillfully reproduce and known dependencies between large and small scales to infer changes in the local (small-scale) climate whereas the latter is about modifying individual grid-box values (below the minimum skillful scales) to match the statistical properties of observations from similar locations.

Rely: We agree that the simulations of coarse scale climate models (GCM and RCM)

are biased, which will be noted in the revised paper.

Regarding the SDMs used in this study, we follow the SDM classification made by Maraun et al. (2010) who classified SDMs into three groups of PP, MOS, and WGs. In that classification, the SDMs 1-3 in this study are classified in the MOS group and the SDM4 in the WGs group. In these methods, "the change factor is calculated for the grid box containing the location of the weather station of interest" (paragraph 81 in Maraun et al., 2010).

L100: I found the section difficult to follow and at times there seemed to be too little information. A rule of thumb is it is annoying to read a paper where you need to read another paper before you understand what it does, and the reader should be able to re-produce the results based on a detailed 'recipe' explaining the method. Reproducibility is an important issue.

Rely: To avoid a lengthy paper, the statistical downscaling methods were briefly ex-plained. We agree with the reviewer that all necessary information to reproduce the results and understand the full context of how to interpret our results should be in-cluded. The statistical downscaling methods will thus be explained in more details.

L101: State that the four methods are explained in more detail below. Also, I'm not sure why there are four methods - what is the merit of each? Sure, I get it that they may give different results, but so what?

Rely: The four SDMs were selected based on their complexity and the way they treat dry spells. Most prominently, each method has a different take on the downscaling of dry spells. This study aims at examining the influence of these factors in the statistical downscaling using four methods which are different in methodology and complexity. While SDM1 and SDM2 are considered simple methods that do not modify dry spells in downscaling, the SDM3 and SDM4 are more advanced methods that adjust dry spells. SDM1 applies a bias correction to the selected statistics, whereas the three other SDMs return a modified precipitation time series. The first method utilizes a direct downscaling strategy by applying the relative change factors directly to the dry spell related research indicators. The other methods opt for an indirect downscaling strategy towards dry spells by integrating the changes in dry days, which are downscaled directly into a coherent time series. For this, SDM2 solely relies on the temporal (precipitation) structure present in the GCM time series. SDM3 on the other hand is expected to actively favor clustering of dry days. Lastly, SDM4 makes use of a probability distribution to sample dry events from. While the precipitation change factor methods (SDMs 1-3) assume independency between successive wet days and apply changes at the daily time scale which can be problematic when successive wet days are part of a longer lasting event, the weather generator (SDM4) identifies precipitation events and applies the same change factor to all precipitation within that event.

In the next version of the manuscript, we will explain the four methods in more details, justify their choice and present their merit.

L107: Insufficient information about the BC methods?

Rely: The BC method as well as the other statistical downscaling methods will be explained in more details in the revised version.

L115: A change factor of mean cannot really be classified as a downscaling method as it does not involve different spatial scales.

Rely: Based on the calibration data, a change factor of mean is classified as a model output statistics (MOS) method (Maraun et al., 2010; Sunyer et al., 2015). It is considered as the simplest MOS method as mentioned in paragraph 66 in Maraun et al. (2010).

L180: Perhaps an illustration will make it easier to follow.

Rely: A figure will be added for a better understanding of the methodology.

L199: Why not downscale the parameters of the pdf for the dry-spell duration? If there is a probability p that it rains on a specific day, then the pdf for the duration of

dry spells ought to follow a binomial distribution with one parameter p = 1/mean(spell lengths). This approach has been tried for heatwaves in India (DOI:10.5194/ascmo-4-37-2018). Because the statistics of spells (duration of events) tend to follow the binomial distribution, it is advisable to include the mean duration - not just its median (which is usually not a quantity that corresponds to the parameters of a pdf).

Rely: - Thank you for the suggesting this interesting method. In this study, two groups of statistical downscaling methods (simple and advanced) were selected to downscale three variables of dry day frequency, dry spell duration and total precipitation. Although the suggested method would effectively downscale dry spell duration, it would not, however, work for the other two variables (total precipitation and dry day frequency). So, the method would be suggested as an interesting downscaling method for future research on the duration of dry spells.

- We agree that dry spell durations follow a binomial distribution which has previously been successfully fitted to the distribution of wet and dry spells in different parts of the world (Wilby et al., 1998; Semenov et al., 1998; Wilks, 1999; Mathlouthi & Lebdi, 2009). In the next version of the paper, the median duration of dry spells would be replaced with the mean duration.

L200-206: There is a subtle but important difference between a rain gauge measurement (a point measurement) and gridbox values from GCMs (area mean estimates), which also has implications for threshold values. This is also expected to be affected by different spatial resolutions (Table 1). For a coarser model, local 'convective' would be smudged out over a larger area (I guess it is parameterised) and expected to have a different statistical characteristic to a very tiny spatial sample (rain gauge measurements with diameters of the order of centimetres). This in itself is a justification for bias adjustment, but nevertheless makes it difficult to compare dry and wet days in models with different spatial resolution.

Rely: The local processes such as convection that cannot be resolved in horizontal grid

spacing of GCMs are parameterized, which is a source of large bias and uncertainty in the simulations. The bias is, therefore, corrected for local impact assessments. A comparison between convection-permitting and convection-parameterized models would lead to a difference in dry spell changes because of a better representation of diurnal convection and propagating systems in convection-permitting models (Kendon et al., 2019). However, at the spatial resolution of GCMs at which local processes are parameterized, the spatial resolution difference is less important compared to the contributions of other factors to the differences across simulations and projections of GCMs such as other differences in the design of climate models, e.g., type of parameterization scheme (Foley, 2010), a lake of knowledge of an unknown future (even without climate change) as well as inadequate theoretical understanding of climate system processes (Knutti et al., 2008; Knutti and Sedláček, 2013) such as climate feedbacks (Knutti and Hegerl, 2008; Collins et al., 2011), carbon cycle processes (Friedlingstein et al., 2006; Booth et al., 2012), boundary-layer, gravity wave drag and its susceptible interplay with large-scale dynamics (Shepherd, 2014) and climate variability (Hegerl et al., 2007).

For the data used in this study, no relation was found between the spatial resolutions of the CMIP6 GCMs and their simulated historical dry day frequencies (p-value=0.2219) and between the spatial resolutions and projected dry day frequency changes of GCMs (p-value=0.1772).

The same threshold value of 1 mm is therefore used for climate change impact assessments on dry spell durations using climate models with a wide range of spatial resolutions; convection-permitting models (e.g., Kendon et al. [2019]), regional climate models (e.g., WRF RCM in Han et al. [2019], EURO‐CORDEX RCMs in Dosio [2016]; Med-CORDEX RCMs in Raymond et al. [2018]; WAS-CORDEX RCMs in Tabari and Willems [2018]; CORDEX-Africa RCMs in Dosio and Panitz [2016]) and global climate models (e.g., CMIP5 GCMs in Sillmann et al. [2013] and Giorgi et al. [2019]; CMIP6 GCMs in Kim et al. [2020]).

L222: Pmax is the max monthly maximum precipitation or the mean monthly maximum?

Rely: Pmax refers to mean monthly maximum precipitation. It will be clarified in the text.

L235: Why is a two-tailed test used here?

Rely: This is because S2N can be either positive or negative as the signal can be either positive or negative (see Tables 5 and 6), but the noise is always positive.

L237: It is important with a proper evaluation of the methods involving e.g. cross-validation and historical data, which seems to be missing. E.g., can the methods reproduce historical changes/variations in dry-spell lengths?

Rely: We agree that assessing the validity of the SDMs is important. In the next version of the manuscript, we will present the calibration results of the SDMs.

L253: 'medium' should be 'median'? It's better to use the mean because the statistics of dry-spell length is expected to follow a distribution that is not too different to the binomial distribution for which the mean is connected to its only parameter.

Rely: We agree to replace median with mean.

L258: Define 'DSL'.

Rely: DSL refers to median very long dry spell length as shown in Table 3. VLDS will be defined in Table 3. DSL will also be defined in the text.

In addition, median very long dry spell length will be replaced with mean very long dry spell length.

L269: It's annoying to stumble across acronyms like 'RI' which I then need to scroll up to remind me of what it stands for. The same goes with the other acronyms. I think it's a bad habit (and sloppy) to use many acronyms and abbreviations and if there is no need to do so, it's generally better to spell out the words to make it easier for the reader

and ensure that (s)he focuses on the message rather than trying to decipher the text. The punishment is of course revisions.

Rely: We agree with the reviewer that the use of many acronyms and abbreviations especially the non-standard ones would be confusing for the reader. To improve the readability of the manuscript, acronyms and abbreviations will be written in the full term as much as possible and acronyms and abbreviations will be used only when absolutely necessary (e.g., space limitation on plots).

I think that the ensemble used here is too small to make any judgement about future climate due to pronounced stochastic local variability on decadal scales (Deser et al., 2012) and the law of small numbers. This stochastic nature should call for the use of probabilistic projections. Furthermore, I would not call the efforts discussed here 'downscaling' but they are more along the line of 'prediction' or 'bias correction' in my opinion (there is no scaling dependencies being utilised between large and small). It can be shown that the large-scale circulation (SLP anomalies) tend to determine whether there is rain or no rain for a location, and downscaling would involve using the large-scale circulation as a means to infer the wet-day frequency and maybe also dry-spell lengths.

Rely: We agree with the reviewer that small samples subject to "the law of small numbers" are susceptible to the presence of strong random statistical fluctuations and can provide misleading results (Benestad et al., 2017a, b). Our climate model ensemble includes 38 runs from 15 GCM for each of the four considered RCP-SSP scenario (SSP1-2.6, SSP2-4.5, SSP3-7.0 and SSP5-8.5). Checking the ESGF website shows that 14 additional CMIP6 GCMs are available, which will be added to the previous ensemble. This makes the total ensemble sample size equal to 53 for each RCP-SSP scenario (simulations from a 29-member ensemble of the CMIP6 GCMs plus 24 extra simulations from the CanESM5 GCM ensemble), which is large enough to limit the effect of the random sampling fluctuation. The law of small numbers and the risk of using a limited sample size will also be noted by citing the relevant references.

As mentioned in paragraph 63 in Maraun et al. (2010), "Depending on the type of simulations used for MOS calibration the predictors can either be simulated precipitation time series or properties of the simulated intensity distribution. Similarly, predictands can either be local precipitation series or properties of the local-scale intensity distribution." In WGs, the change factors are used to modify observations for a future climate. "Once these change factors are calculated, no large-scale drivers are needed to generate weather time series." (paragraph 81 in Maraun et al., 2010).

L380: Need to define 'accurately' and provide evidence for the statement.

Rely: The validation results of the SDMs will be presented in the next version.

USED REFERENCES:

Benestad, R., Parding, K., Dobler, A., & Mezghani, A. (2017a). A strategy to effectively make use of large volumes of climate data for climate change adaptation. Climate Services, 6, 48-54.

Benestad, R., et al. (2017b). New vigour involving statisticians to overcome ensemble fatigue. Nature Climate Change, 7(10), 697-703.

Booth, B. B., Dunstone, N. J., Halloran, P. R., Andrews, T., & Bellouin, N. (2012). Aerosols implicated as a prime driver of twentieth-century North Atlantic climate variability. Nature, 484(7393), 228-232.

Collins, M., Booth, B. B., Bhaskaran, B., Harris, G. R., Murphy, J. M., Sexton, D. M., & Webb, M. J. (2011). Climate model errors, feedbacks and forcings: a comparison of perturbed physics and multi-model ensembles. Climate Dynamics, 36(9-10), 1737-1766.

Dosio, A. (2016). Projections of climate change indices of temperature and precipitation from an ensemble of bias‐adjusted high‐resolution EURO‐CORDEX regional climate models. Journal of Geophysical Research: Atmospheres, 121(10), 5488-5511.

Dosio, A., & Panitz, H. J. (2016). Climate change projections for CORDEX-Africa with COSMO-CLM regional climate model and differences with the driving global climate models. Climate Dynamics, 46(5-6), 1599-1625.

Friedlingstein, P., Cox, P., Betts, R., Bopp, L., von Bloh, W., Brovkin, V., ... & Bala, G. (2006). Climate–carbon cycle feedback analysis: results from the C4MIP model intercomparison. Journal of Climate, 19(14), 3337-3353.

Giorgi, F., Raffaele, F., & Coppola, E. (2019). The response of precipitation characteristics to global warming from climate projections. Earth System Dynamics, 10(1), 73-89.

Han, F., Cook, K. H., & Vizy, E. K. (2019). Changes in intense rainfall events and dry periods across Africa in the twenty-first century. Climate Dynamics, 53(5-6), 2757-2777.

Hegerl, G. C., et al. (2007). Understanding and attributing climate change. Climate Change 2007: The Physical Science Basis, S. Solomon et al., Eds., Cambridge University Press.

Kendon, E. J., Stratton, R. A., Tucker, S., Marsham, J. H., Berthou, S., Rowell, D. P., & Senior, C. A. (2019). Enhanced future changes in wet and dry extremes over Africa at convection-permitting scale. Nature Communications, 10(1), 1-14.

Kim, Y. H., Min, S. K., Zhang, X., Sillmann, J., & Sandstad, M. (2020). Evaluation of the CMIP6 multi-model ensemble for climate extreme indices. Weather and Climate Extremes, 29, 100269.

Knutti, R., & Hegerl, G. C. (2008). The equilibrium sensitivity of the Earth's temperature to radiation changes. Nature Geoscience, 1(11), 735-743.

Knutti, R., & Sedláček, J. (2013). Robustness and uncertainties in the new CMIP5 climate model projections. Nature Climate Change, 3(4), 369-373.

[Figure]

Knutti, R., Allen, M. R., Friedlingstein, P., Gregory, J. M., Hegerl, G. C., Meehl, G. A., ... & Stocker, T. F. (2008). A review of uncertainties in global temperature projections over the twenty-first century. Journal of Climate, 21(11), 2651-2663.

Mathlouthi, M., & Lebdi, F. (2009). Statistical analysis of dry events in a northern Tunisian basin. Hydrological Sciences Journal, 54(3), 422-455.

Raymond, F., Drobinski, P., Ullmann, A., & Camberlin, P. (2018). Extreme dry spells over the Mediterranean Basin during the wet season: Assessment of HyMeX/Med‐CORDEX regional climate simulations (1979–2009). International Journal of Climatology, 38(7), 3090-3105.

Semenov, M. A., Brooks, R. J., Barrow, E. M., & Richardson, C. W. (1998). Comparison of the WGEN and LARS-WG stochastic weather generators for diverse climates. Climate Research, 10(2), 95-107.

Shepherd, T. G. (2014). Atmospheric circulation as a source of uncertainty in climate change projections. Nature Geoscience, 7(10), 703-708.

Sillmann, J., Kharin, V. V., Zhang, X., Zwiers, F. W., & Bronaugh, D. (2013). Climate extremes indices in the CMIP5 multimodel ensemble: Part 1. Model evaluation in the present climate. Journal of Geophysical Research: Atmospheres, 118(4), 1716-1733.

Tabari, H., & Willems, P. (2018). More prolonged droughts by the end of the century in the Middle East. Environmental Research Letters, 13(10), 104005.

Wilby, R. L., Wigley, T. M. L., Conway, D., Jones, P. D., Hewitson, B. C., Main, J., & Wilks, D. S. (1998). Statistical downscaling of general circulation model output: A comparison of methods. Water Resources Research, 34(11), 2995-3008.

Wilks, D. S. (1999). Interannual variability and extreme-value characteristics of several stochastic daily precipitation models. Agricultural and Forest Meteorology, 93(3), 153-169.

---

## Author Response (AR1)

**Comparison of statistical downscaling methods for climate change impact analysis on precipitation-driven drought**

We appreciate all the useful comments and suggestions provided by the anonymous reviewers. The changes to the text, which address the different comments, were highlighted in **BLUE COLOR** in the revised manuscript.

**Referee #1**

HESS-2020-506 Title: Comparison of statistical downscaling methods for climate change impact analysis on drought
Authors: Hossein Tabari, Daan Buekenhout, Patrick Willems

GENERAL COMMENT
The paper presents a comparative analysis of four (4) Statistical Downscaling Methods (SDMs), namely, Bias Correction (BC), Change Factor of Mean (CFM), Quantile Perturbation (QP) and Event Based Weather Generator (EBWG) to assess climate change impact on drought by the end of the 21st century (2071-2100) relative to a baseline period of 1971-2000. The SDMs were applied to downscale daily precipitation from 14-member ensemble of CMIP6 GCM at the Uccle weather station in Belgium for four future scenarios, namely, SSP1-2.6, SSP2-4.5, SSP3-7.0 and SSP5-8.5. Various drought indices have been calculated and used in the comparison of SDMs' results for the future period with the drought indices estimated from the observed precipitation for the baseline period.
The paper is well organized, written and comprehensive. However, there are a few points that should be clarified and addressed. Overall, the paper merits publication in the HESS after the moderate comments are properly addressed.

*REPLY: We thank the reviewer for the encouraging assessment and helpful comments.*

SPECIFIC COMMENTS
1) Title: The title of the paper should include the word "precipitation". Precipitation is the variable downscaled from GCMs in the paper. Additionally, drought phenomenon is affected by other meteorological variables, for example temperature, evapotranspiration and others. It should be clear from the title that the work presented in the paper deals with the downscaling of precipitation for the estimation of climate change impacts on droughts.

*REPLY: We agree with the reviewer that the title of the paper should show that precipitation and its lack are investigated in this study. The title was revised as "Comparison of statistical downscaling methods for climate change impact analysis on precipitation-driven drought".*

2) Abstract: It should be made clear in the Abstract that the downscaling exercise was made for the weather station of Uccle located in Belgium. This information is missing from the stand-alone abstract of the paper.

*REPLY: The weather station name was added to the abstract.*

3) It is usual to calibrate/validate the SDMs during the historical base period (1971-2000 in the paper) and, then, apply them for the future period(s). The authors although mentioned that the methods have been calibrated using the observed precipitation at the Uccle weather station, they do not present any results (i.e. statistics, graphs) about the calibration of the SDMs. The presentation of the calibration results are necessary to assess the validity of the downscaling methods before using them to the future periods.

*REPLY: We agree that assessing the validity of the statistical downscaling methods is important. We evaluated the skill of the statistical downscaling methods using two cross-validation methods. The validation procedure was explained in section 2.3. The validation results are shown in new figures 1 and 2 and interpreted in section 3 (lines 317-325). The validation methods were also explained in section 2.3.*

MINOR COMMENTS

4) Lines 46-48. It is written "Precipitation and the number of wet days were found to increase during summer and to decrease during winter, while evapotranspiration was found to increase for both seasons. This suggests drier summers and wetter winters." This statement is quite vague and needs further explanation on the ratio of precipitation and evapotranspiration to generate drier summers and wetter winters.

*REPLY: The difference between precipitation and evapotranspiration was used as an indicator of drought and water availability in that study. The sentence was revised by clarifying this issue.*

5) Line 130. The equation of m should be written better. The equal sign is not shown properly.

*REPLY: A space was added before the equal sign.*

6) Table 3. What is WLDS? Although the term is presented in the text of the paper, it should be written in full. Tables are stand-alone elements of a paper.

*REPLY: VLDS refers to very long dry spell. It was written in full in the table. The abbreviations in the other tables were also written in full.*

7) Figure 2. The color bars and the reference lines should be explained.

*REPLY: Sorry for the missing legend. The legend was added to the figure (Fig. 6 in the revised paper).*
=======================================================================

**Referee #2**

In general, I found this manuscript difficult to follow. If I weren't supposed to review it, I would have stopped reading it fairly quickly. This is not so much a question about the English but the way the material is structured and that it presumes that the reader already is familiar with the work. I also think I have some misgivings about the results and have a few issues with the nomenclature. There is no proper evaluation of the methods, and I think the authors could give more background and discuss their efforts in the context of other relevant papers.

*REPLY: We thank the reviewer for the critical assessment and constructive feedback.*
*The major improvements made in the revised paper are:*

- *We enlarged the ensemble sample size to limit the effect of the random sampling fluctuation, making the total ensemble sample size equal to 52 for each RCP-SSP scenario (simulations from a 28-member ensemble of the CMIP6 GCMs plus 24 extra simulations from the CanESM5 GCM ensemble). In this regard, 'the law of small numbers' and the risk of using a limited sample size were noted by citing the relevant references (lines 99-100).*
- *We explained the four methods in more details and justified their choice in the revised paper.*
- *We evaluated the skill of the statistical downscaling methods using two cross-validation methods to show how they can reproduce dry day frequency, dry spell duration and total precipitation. The validation procedure was explained in section 2.3. The validation results are shown in new figures 1 and 2 and interpreted in section 3 (lines 317-325). The validation methods were also explained in section 2.3.*
- *To improve the readability of the manuscript, all acronyms and abbreviations except the four downscaling methods were written in the full term in the revised paper. The English of the paper was also improved.*
- *The results of the study were discussed in the context of other relevant studies, and the literature review was modified.*

L.40-48: It may be of interest to note that the occurrence of droughts also may be a consequence of a reduction in the global area with 24-hr rainfall (DOI: 10.1088/1748-9326/aab375).

*REPLY: We appreciate the reviewer for bringing this point to our attention. The drought as a result of a decrease in daily precipitation area was noted in the revised paper.*

L.66: The statement 'As there is no single best downscaling method' may be correct generally for a range of situations, but for specific and limited problems, there may be some method that is superior and provides the most reliable answer.

*REPLY: The sentence refers to the lack of a single best downscaling method for all applications and regions. We agree with the reviewer that some methods are superior for specific applications. That is why we need to evaluate different statistical downscaling methods to select the optimal one for each application based on the information needs (e.g., desired spatial and temporal resolutions) and on available resources (data, expertise, computing resources and time-frames).*
    *The sentence was reformulated in the revised paper to clarify this point.*

L.73/79: ensemble sizes of 14 or 25 members are on the low end for giving robust information, due to pronounced and stochastic/chaotic decadal variations (Deser et al., 2012) and 'the law of small numbers'. If the ensemble is to be regarded as a statistical sample, then it typically needs more than 30 data points to get results that are not heavily affected by random sampling fluctuation, and preferably more than 100 members to get more robust statistics.

*REPLY: We agree with the reviewer that small samples subject to "the law of small numbers" are susceptible to the presence of strong random statistical fluctuations and can provide misleading results (Benestad et al., 2017a, b). Our initial climate model ensemble includes 38 runs from 15 GCM for each of the four considered RCP-SSP scenario (SSP1-2.6, SSP2-4.5, SSP3-7.0 and SSP5-8.5). Thirteen additional GCMs were added to the previous ensemble. This makes the total ensemble sample size equal to 52 for each RCP-SSP scenario (simulations from a 28-member ensemble of the CMIP6 GCMs plus 24 extra simulations from the CanESM5 GCM ensemble), which is large enough to limit the effect of the random sampling fluctuation. The law of small numbers and the risk of using a limited sample size were noted by citing the relevant references (lines 99-100).*

L83: If 'precipitation time series produced by GCMs as sole predictor' also is used for model calibration, then there seems to be a problem because the GCMs are not synchronised with the observations in the real world.

*REPLY: Rummukainen (1997) classified SDMs into two groups of perfect prognosis (PP) and model output statistics (MOS) based on the information used for downscaling, and Wilby and Wigely (1997) and Fowler et al. (2007) classified them into three groups of regression methods, weather type approaches, and stochastic weather generators (WGs) based on the relationship used to connect large and local scales. Maraun et al. (2010) combined these two types of the SDM classification and came up with three categories of PP, MOS, and WGs. According to the last classification, the SDMs 1-3 in this study are classified as MOS methods and the SDM 4 as a WG.*

*As mentioned in paragraph 63 in Maraun et al. (2010), "Depending on the type of simulations used for MOS calibration the predictors can either be simulated precipitation time series or properties of the simulated intensity distribution. Similarly, predictands can either be local precipitation series or properties of the local-scale intensity distribution."*

*In WGs, the change factors are used to modify observations for a future climate. "Once these change factors are calculated, no large-scale drivers are needed to generate weather time series." (paragraph 81 in Maraun et al., 2010).*

L92: Both GCMs and RCMs have a minimum skillful scale (DOI: 10.2151/jmsj.2015-042) which implies that single grid box values should not be used as a predictor. The difference between downscaling and bias adjustment is that the former utilises the scales that the models are able to skillfully reproduce and known dependencies between large and small scales to infer changes in the local (small-scale) climate whereas the latter is about modifying individual grid-box values (below the minimum skillful scales) to match the statistical properties of observations from similar locations.

*REPLY: We agree that the simulations of coarse scale climate models (GCM and RCM) are biased, which was noted in the lines 51-53 of the revised paper.*

*Regarding the SDMs used in this study, we follow the SDM classification made by Maraun et al. (2010) who classified SDMs into three groups of PP, MOS, and WGs. In that classification, the SDMs 1-3 in this study are classified in the MOS group and the SDM4 in the WGs group. In these methods, "the change factor is calculated for the grid box containing the location of the weather station of interest" (paragraph 81 in Maraun et al., 2010).*

L100: I found the section difficult to follow and at times there seemed to be too little information. A rule of thumb is it is annoying to read a paper where you need to read another paper before you understand what it does, and the reader should be able to reproduce the results based on a detailed 'recipe' explaining the method. Reproducibility is an important issue.

*REPLY: To avoid a lengthy paper, the statistical downscaling methods were briefly explained in the initial version of the paper. We agree with the reviewer that all necessary information to reproduce the results and understand the full context of how to interpret our results should be included. The statistical downscaling methods were explained in more details in the revised paper (L110-257).*

L101: State that the four methods are explained in more detail below. Also, I'm not sure why there are four methods - what is the merit of each? Sure, I get it that they may give different results, but so what?

*REPLY: The four statistical downscaling methods were selected for this study based on their complexity and the way they treat dry spells. Each method has a different take on the downscaling of dry spells. This study aims at examining the influence of these factors in the statistical downscaling using four methods which are different in methodology and complexity. While BC and CFM are considered simple and computationally fast and straightforward methods that do not modify dry spells in downscaling, QP and WB are more advanced methods that adjust dry spells. BC applies a bias correction to the selected statistics, whereas the other three downscaling methods return a modified precipitation time series. BC utilizes a direct downscaling strategy by applying the relative change factors directly to the dry spell related research indicators. The other three methods opt for an indirect downscaling strategy towards dry spells by integrating the changes in dry days, which are downscaled directly into a coherent time series. For this, CFM solely relies on the temporal (precipitation) structure present in the GCM time series. QP on the other hand is expected to actively favour clustering of dry days. Lastly, WB makes use of a probability distribution to sample dry events from. While the precipitation change factor methods (BC, CFM and QP) assume independency between successive wet days and apply changes at the daily time scale, which can be problematic when successive wet days are part of a longer lasting event, WG identifies precipitation events and applies the same change factor to all precipitation within that event.*

*The above explanation is present in lines 111-124 of the revised paper. In addition, we explained the four methods in more details (L125-257).*

L107: Insufficient information about the BC methods?

*REPLY: The statistical downscaling methods were explained in more details in the revised paper (L111-257).*

L115: A change factor of mean cannot really be classified as a downscaling method as it does not involve different spatial scales.

*REPLY: Based on the calibration data, a change factor of mean is classified as a model output statistics (MOS) method (Maraun et al., 2010; Sunyer et al., 2015). It is considered as the simplest MOS method as mentioned in paragraph 66 in Maraun et al. (2010).*

L180: Perhaps an illustration will make it easier to follow.

*REPLY: Relevant equations were added for a better understanding of the method.*

L199: Why not downscale the parameters of the pdf for the dry-spell duration? If there is a probability p that it rains on a specific day, then the pdf for the duration of dry spells ought to follow a binomial distribution with one parameter p = 1/mean(spell lengths). This approach has been tried for heatwaves in India (DOI:10.5194/ascmo-4-37-2018). Because the statistics of spells (duration of events) tend to follow the binomial distribution, it is advisable to include the mean duration - not just its median (which is usually not a quantity that corresponds to the parameters of a pdf).

*REPLY:*

*- Thank you for the suggesting this interesting method. In this study, two groups of statistical downscaling methods (simple and advanced) were selected to downscale three variables of dry day frequency, dry spell duration and total precipitation. Although the suggested method would effectively downscale dry spell duration, it would not, however, work for the other two variables (total precipitation and dry day frequency). So, the method would be suggested as an interesting downscaling method for future research on the duration of dry spells.*

*- We agree that dry spell durations follow a binomial distribution which has previously been successfully fitted to the distribution of wet and dry spells in different parts of the world (Wilby et al., 1998; Semenov et al., 1998; Wilks, 1999; Mathlouthi & Lebdi, 2009).*

*In the revised paper, the median duration of dry spells was replaced with the mean duration.*

L200-206: There is a subtle but important difference between a rain gauge measurement (a point measurement) and gridbox values from GCMs (area mean estimates), which also has implications for threshold values. This is also expected to be affected by different spatial resolutions (Table 1). For a coarser model, local 'convective' would be smudged out over a larger area (I guess it is parameterised) and expected to have a different statistical characteristic to a very tiny spatial sample (rain gauge measurements with diameters of the order of centimetres). This in itself is a justification for bias adjustment, but nevertheless makes it difficult to compare dry and wet days in models with different spatial resolution.

*REPLY: The local processes such as convection that cannot be resolved in horizontal grid spacing of GCMs are parameterized, which is a source of large bias and uncertainty in the simulations. The bias is, therefore, corrected for local impact assessments. A comparison between convection-permitting and convection-parameterized models would*

*lead to a difference in dry spell changes because of a better representation of diurnal convection and propagating systems in convection-permitting models (Kendon et al., 2019). However, at the spatial resolution of GCMs at which local processes are parameterized, the spatial resolution difference is less important compared to the contributions of other factors to the differences across simulations and projections of GCMs such as other differences in the design of climate models, e.g., type of parameterization scheme (Foley, 2010), a lake of knowledge of an unknown future (even without climate change) as well as inadequate theoretical understanding of climate system processes (Knutti et al., 2008; Knutti and Sedláček, 2013) such as climate feedbacks (Knutti and Hegerl, 2008; Collins et al., 2011), carbon cycle processes (Friedlingstein et al., 2006; Booth et al., 2012), boundary-layer, gravity wave drag and its susceptible interplay with large-scale dynamics (Shepherd, 2014) and climate variability (Hegerl et al., 2007).*

*For the data used in this study, no relation was found between the spatial resolutions of the CMIP6 GCMs and their simulated historical dry day frequencies (p-value=0.2219) and between the spatial resolutions and projected dry day frequency changes of GCMs (p-value=0.1772).*

*The same threshold value of 1 mm is therefore used for climate change impact assessments on dry spell durations using climate models with a wide range of spatial resolutions; convection-permitting models (e.g., Kendon et al. [2019]), regional climate models (e.g., WRF RCM in Han et al. [2019], EURO-CORDEX RCMs in Dosio [2016]; Med-CORDEX RCMs in Raymond et al. [2018]; WAS-CORDEX RCMs in Tabari and Willems [2018]; CORDEX-Africa RCMs in Dosio and Panitz [2016]) and global climate models (e.g., CMIP5 GCMs in Sillmann et al. [2013] and Giorgi et al. [2019]; CMIP6 GCMs in Kim et al. [2020]).*

L222: Pmax is the max monthly maximum precipitation or the mean monthly maximum?

*REPLY: Pmax refers to mean monthly maximum precipitation. It is monthly maximum daily precipitation averaged over the 30-year period.*

L235: Why is a two-tailed test used here?

*REPLY: This is because S2N can be either positive or negative as the signal can be either positive or negative (see Tables 5 and 6), but the noise is always positive.*

L237: It is important with a proper evaluation of the methods involving e.g. cross-validation and historical data, which seems to be missing. E.g., can the methods reproduce historical changes/variations in dry-spell lengths?

*REPLY: We evaluated the skill of the statistical downscaling methods using two cross-validation methods to show how they can reproduce dry day frequency, dry spell duration and total precipitation. The validation procedure was explained in section 2.3. The validation results are shown in new figures 1 and 2 and interpreted in section 3 (lines 317-325). The validation methods were also explained in section 2.3.*

L253: 'medium' should be 'median'? It's better to use the mean because the statistics of dry-spell length is expected to follow a distribution that is not too different to the binomial distribution for which the mean is connected to its only parameter.

*REPLY: We agree to replace median with mean.*

L258: Define 'DSL'.

*REPLY: DSL refers to median very long dry spell length as shown in Table 3. VLDS was written in full in Table 3 and in the text. In addition, median very long dry spell length was replaced with mean very long dry spell length.*

L269: It's annoying to stumble across acronyms like 'RI' which I then need to scroll up to remind me of what it stands for. The same goes with the other acronyms. I think it's a bad habit (and sloppy) to use many acronyms and abbreviations and if there is no need to do so, it's generally better to spell out the words to make it easier for the reader and ensure that (s)he focuses on the message rather than trying to decipher the text. The punishment is of course revisions.

*REPLY: We agree with the reviewer that the use of many acronyms and abbreviations especially the non-standard ones would be confusing for the reader. To improve the readability of the manuscript, all acronyms and abbreviations except the four downscaling methods were written in the full term in the revised paper.*

I think that the ensemble used here is too small to make any judgement about future climate due to pronounced stochastic local variability on decadal scales (Deser et al., 2012) and the law of small numbers. This stochastic nature should call for the use of probabilistic projections. Furthermore, I would not call the efforts discussed here 'downscaling' but they are more along the line of 'prediction' or 'bias correction' in my opinion (there is no scaling dependencies being utilised between large and small). It can be shown that the large-scale circulation (SLP anomalies) tend to determine whether there is rain or no rain for a location, and downscaling would involve using the large-scale circulation as a means to infer the wet-day frequency and maybe also dry-spell lengths.

*REPLY: We agree with the reviewer that small samples subject to "the law of small numbers" are susceptible to the presence of strong random statistical fluctuations and can provide misleading results (Benestad et al., 2017a, b). Our initial climate model ensemble includes 38 runs from 15 GCM for each of the four considered RCP-SSP scenario (SSP1-2.6, SSP2-4.5, SSP3-7.0 and SSP5-8.5). Thirteen additional GCMs were added to the previous ensemble. This makes the total ensemble sample size equal to 52 for each RCP-SSP scenario (simulations from a 28-member ensemble of the CMIP6 GCMs plus 24 extra simulations from the CanESM5 GCM ensemble), which is large enough to limit the effect of the random sampling fluctuation. The law of small numbers and the risk of using a limited sample size were noted by citing the relevant references (lines 99-100).*

*As mentioned in paragraph 63 in Maraun et al. (2010), "Depending on the type of simulations used for MOS calibration the predictors can either be simulated precipitation time series or properties of the simulated intensity distribution. Similarly, predictands can either be local precipitation series or properties of the local-scale intensity distribution." In WGs, the change factors are used to modify observations for a future climate. "Once these change factors are calculated, no large-scale drivers are needed to generate weather time series." (paragraph 81 in Maraun et al., 2010).*

L380: Need to define 'accurately' and provide evidence for the statement.

*REPLY: We evaluated the skill of the statistical downscaling methods using two cross-validation methods to show how they can reproduce dry day frequency, dry spell duration and total precipitation. The validation procedure was explained in section 2.3. The validation results are shown in new figures 1 and 2 and interpreted in section 3 (lines 317-325). The validation methods were also explained in section 2.3.*

*Used references:*

*Benestad, R., et al. (2017b). New vigour involving statisticians to overcome ensemble fatigue. Nature Climate Change, 7(10), 697-703.*

*Benestad, R., Parding, K., Dobler, A., & Mezghani, A. (2017a). A strategy to effectively make use of large volumes of climate data for climate change adaptation. Climate Services, 6, 48-54.*

*Booth, B. B., Dunstone, N. J., Halloran, P. R., Andrews, T., & Bellouin, N. (2012). Aerosols implicated as a prime driver of twentieth-century North Atlantic climate variability. Nature, 484(7393), 228-232.*

*Collins, M., Booth, B. B., Bhaskaran, B., Harris, G. R., Murphy, J. M., Sexton, D. M., & Webb, M. J. (2011). Climate model errors, feedbacks and forcings: a comparison of perturbed physics and multi-model ensembles. Climate Dynamics, 36(9-10), 1737-1766.*

*Dosio, A. (2016). Projections of climate change indices of temperature and precipitation from an ensemble of bias-adjusted high-resolution EURO-CORDEX regional climate models. Journal of Geophysical Research: Atmospheres, 121(10), 5488-5511.*

*Dosio, A., & Panitz, H. J. (2016). Climate change projections for CORDEX-Africa with COSMO-CLM regional climate model and differences with the driving global climate models. Climate Dynamics, 46(5-6), 1599-1625.*

*Friedlingstein, P., Cox, P., Betts, R., Bopp, L., von Bloh, W., Brovkin, V., ... & Bala, G. (2006). Climate–carbon cycle feedback analysis: results from the C4MIP model intercomparison. Journal of Climate, 19(14), 3337-3353.*

*Giorgi, F., Raffaele, F., & Coppola, E. (2019). The response of precipitation characteristics to global warming from climate projections. Earth System Dynamics, 10(1), 73-89.*

*Han, F., Cook, K. H., & Vizy, E. K. (2019). Changes in intense rainfall events and dry periods across Africa in the twenty-first century. Climate Dynamics, 53(5-6), 2757-2777.*

*Hegerl, G. C., et al. (2007). Understanding and attributing climate change. Climate Change 2007: The Physical Science Basis, S. Solomon et al., Eds., Cambridge University Press.*

*Kendon, E. J., Stratton, R. A., Tucker, S., Marsham, J. H., Berthou, S., Rowell, D. P., & Senior, C. A. (2019). Enhanced future changes in wet and dry extremes over Africa at convection-permitting scale. Nature Communications, 10(1), 1-14.*

*Kim, Y. H., Min, S. K., Zhang, X., Sillmann, J., & Sandstad, M. (2020). Evaluation of the CMIP6 multi-model ensemble for climate extreme indices. Weather and Climate Extremes, 29, 100269.*

*Knutti, R., & Hegerl, G. C. (2008). The equilibrium sensitivity of the Earth's temperature to radiation changes. Nature Geoscience, 1(11), 735-743.*

Knutti, R., & Sedláček, J. (2013). Robustness and uncertainties in the new CMIP5 climate model projections. *Nature Climate Change, 3(4),* 369-373.

Knutti, R., Allen, M. R., Friedlingstein, P., Gregory, J. M., Hegerl, G. C., Meehl, G. A., ... & Stocker, T. F. (2008). A review of uncertainties in global temperature projections over the twenty-first century. *Journal of Climate, 21(11),* 2651-2663.

Mathlouthi, M., & Lebdi, F. (2009). Statistical analysis of dry events in a northern Tunisian basin. *Hydrological Sciences Journal, 54(3),* 422-455.

Raymond, F., Drobinski, P., Ullmann, A., & Camberlin, P. (2018). Extreme dry spells over the Mediterranean Basin during the wet season: Assessment of HyMeX/Med-CORDEX regional climate simulations (1979–2009). *International Journal of Climatology, 38(7),* 3090-3105.

Semenov, M. A., Brooks, R. J., Barrow, E. M., & Richardson, C. W. (1998). Comparison of the WGEN and LARS-WG stochastic weather generators for diverse climates. *Climate Research, 10(2),* 95-107.

Shepherd, T. G. (2014). Atmospheric circulation as a source of uncertainty in climate change projections. *Nature Geoscience, 7(10),* 703-708.

Sillmann, J., Kharin, V. V., Zhang, X., Zwiers, F. W., & Bronaugh, D. (2013). Climate extremes indices in the CMIP5 multimodel ensemble: Part 1. Model evaluation in the present climate. *Journal of Geophysical Research: Atmospheres, 118(4),* 1716-1733.

Tabari, H., & Willems, P. (2018). More prolonged droughts by the end of the century in the Middle East. *Environmental Research Letters, 13(10),* 104005.

Wilby, R. L., Wigley, T. M. L., Conway, D., Jones, P. D., Hewitson, B. C., Main, J., & Wilks, D. S. (1998). Statistical downscaling of general circulation model output: A comparison of methods. *Water Resources Research, 34(11),* 2995-3008.

Wilks, D. S. (1999). Interannual variability and extreme-value characteristics of several stochastic daily precipitation models. *Agricultural and Forest Meteorology, 93(3),* 153-169.

---

## Author Response (AR2)

**Comparison of statistical downscaling methods for climate change impact analysis on precipitation-driven drought**

We appreciate the minor technical comments by the anonymous reviewer. They were all applied in the text.

**Referee #2**

Minor technical corrections:

- check line 213 - I think a verb is missing in the sentence (after the parenthesis).

*REPLY: The sentence was revised. Thanks.*

- check if definitions for 'fi' is provided or if it should be 'cF'.

*REPLY: It is $cf_i$ which was defined in the text.*

- Line 508: should 'Besides' be 'Beside'?

*REPLY: Done.*